# CaliGCL: Calibrated Graph Contrastive Learning via Partitioned Similarity and Consistency Discrimination

**Yuena Lin [1,†], Hao Wei [1,†], Haichun Cai [2], Bohang Sun [1],**
**Tao Yang [3], Zhen Yang [1,·], & Gengyu Lyu [1,∗]**
[1] College of Computer Science, Beijing University of Technology, Beijing
[2] College of Computer and Data Science, Fuzhou University, Fuzhou
[3] Idealism Beijing Technology Co., Ltd., Beijing
yuenalin@126.com, haowei@emails.bjut.edu.cn, fjsmchc@163.com
sunbohang@bjut.edu.cn, yangtao@ilxzy.cn, yangzhen@bjut.edu.cn
lyugengyu@gmail.com

## Abstract

Graph contrastive learning (GCL) aims to learn self-supervised representations by distinguishing positive and negative sample pairs generated from multiple augmented graph views. Despite showing promising performance, GCL still suffers from two critical biases: (1) **Similarity estimation bias** arises when feature elements that support positive pair alignment are suppressed by conflicting components within the representation, causing truly positive pairs to appear less similar. (2) **Semantic shift bias** occurs when random augmentations alter the underlying semantics of samples, leading to incorrect positive or negative assignments and injecting noise into training. To address these issues, we propose CaliGCL, a GCL model for calibrating the biases by integrating an exponential partitioned similarity measure and a semantics-consistency discriminator. The exponential partitioned similarity computes the similarities among fine-grained partitions obtained through splitting representation vectors and uses exponential scaling to emphasize aligned (positive) partitions while reducing the influence of misaligned (negative) ones. The discriminator dynamically identifies whether augmented sample pairs maintain semantic consistency, enabling correction of misleading contrastive supervision signals. These components jointly reduce biases in similarity estimation and sample pairing, guiding the encoder to learn more robust and semantically meaningful representations. Extensive experiments on multiple benchmarks show that CaliGCL effectively mitigates both types of biases and achieves state-of-the-art performance.

## 1 Introduction

General graph contrastive learning (GCL) follows a pipeline to realize self-supervised representation learning [1, 2, 3], which includes: implement augmentations on raw data to generate multiple views [4], encode the augmented data to produce representations [5], and finally adopt a common contrastive supervision that corresponding samples across different augmented views are positives while others are negatives for contrastive learning [6, 7, 8]. Even though this GCL framework is well recognized, it still incurs **similarity estimation bias** and **semantic shift bias**, which may degrade the graph encoder to generate inferior representations.

The similarity estimation bias arises from the fact that representations produced by the encoder are typically composed of conflicting positively- and negatively-synergistic feature elements, where the

---

[∗]Corresponding author, † equal contributions.

39th Conference on Neural Information Processing Systems (NeurIPS 2025).

positively-synergistic feature elements are those for supporting the alignment of latent positive pairs within representation vectors, while the negatively-synergistic feature elements are those that work against the positively-synergistic ones. When computing the similarity (commonly via inner product) between two representations, these conflicting feature elements would cancel each other out even within positive pairs. This undermines the model ability to identify the latent positive pairs, leading to suboptimal representation learning.

To illustrate how negatively-synergistic features suppress similarity, we conduct a statistical analysis on the ratio between positively-aligned and negatively-aligned partitions under different partition settings in Figure 1, where a partition is positively-aligned if the corresponding feature elements have the positive inner product, otherwise negatively-aligned. Each partition refers to a subset of feature elements within a representation. In the experiment, we follow the GCL framework to train an encoder on Citeseer dataset and divide the generated representations into multiple partitions. With label guidance, we calculate the ratios of positively- and negatively-aligned partitions within positive sample pairs. As the number of partitions increases, the ratio of negatively-aligned partitions grows, revealing the prevalence of conflicting feature elements in the learned representations. This trend suggests that *when computing overall similarity via inner product, multiple negatively-aligned partitions collectively suppress positively-aligned ones, ultimately leading to the similarity estimation bias*.

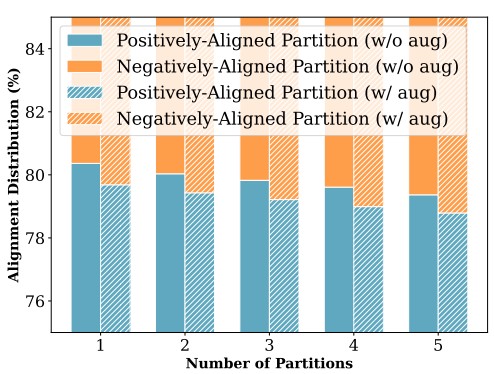

Figure 1: Ratio (alignment distribution) of positively-aligned to negatively-aligned similarity partitions under different partition settings. Shaded histogram bars represent results obtained with graph augmentations, while unshaded ones denote results without augmentations.

Apart from the similarity estimation bias, random graph augmentations in GCL easily flip the relationship between sample pairs, which would result in semantic shift bias. Specifically, graph augmentations as common tools in GCL [9, 10] may alter essential node features or graph topology [11, 12], which potentially flips the relationship between sample pairs: semantically consistent sample pairs may be assigned as negatives, while semantically divergent ones may be incorrectly treated as positives [13]. This introduces noise into contrastive supervision and misguides the training process [14, 15]. Figure 1 illustrates this problem under the condition where the number of partitions is only 1, and the inner product of the partition is equivalent to the inner product of two representation vectors. It could be found that the ratios change when comparing the alignment distributions with and without augmentations, which suggests that augmentations possibly flip the nature of sample pairs (positive ↔ negative). Importantly, *when these flipped pairs are treated as fixed positives or negatives under conventional contrastive supervision, the error will propagate through the training process, causing the encoder to become biased due to semantic shift*.

To calibrate the biases in GCL, we propose a novel graph contrastive model CaliGCL, which integrates an exponential partitioned similarity measure and a semantics-consistency discriminator. The exponential partitioned similarity measure first splits each representation vector into fine-grained partitions and computes the inner product similarity among each partition, then applies an exponential function to shrink the negatively-aligned partition similarities while amplifying the positively-aligned partition similarities, thereby improving the influence of positively-synergistic feature elements and reducing the suppression of negatively-synergistic feature elements to alleviate the similarity estimation bias. The semantics-consistency discriminator [16, 17] takes the sample pair information as input and dynamically predicts relationship consistency scores after augmentation, which are used for calibrating the contrastive supervision to enhance the resilience of GCL and combat semantic shift bias. The two components cooperate to realize debiased GCL and guide the encoder to learn more robust and semantically meaningful representations. Our contributions are summarized as follows:

- We identify two critical biases in the standard GCL framework, namely similarity estimation bias and semantic shift bias, and propose a novel calibrated graph contrastive learning model, CaliGCL, to systematically address them.

- We propose an exponential partitioned similarity measure to restrict the negatively-synergistic features and amplify the positively-synergistic features to alleviate the similarity estimation bias. Besides, a semantics-consistency discriminator is proposed to correct the semantic relationships of augmented sample pairs, effectively debiasing semantic shift.

- Extensive experiments across multiple datasets show that the proposed model consistently surpasses advanced self-supervised graph models, which demonstrate that our model effectively calibrates the biases in GCL and enhances the representation quality.

## 2 Related works

**Problem definition and notations.** In this paper, a graph is denoted as $\mathcal{G} = (\mathcal{V}, \mathcal{E})$, where $\mathcal{V} = \{v_1, v_2, \cdots, v_N\}$ is the node set with $N$ nodes and $\mathcal{E} \subset \mathcal{V} \times \mathcal{V}$ is the edge set. The node attribute matrix and adjacency matrix are denoted as $\boldsymbol{X} \in \mathbb{R}^{N \times F}$ and $\boldsymbol{A} \in \{0, 1\}^{N \times N}$, respectively. We aim to train an $L$-layer graph encoder $f : \mathbb{R}^{N \times F} \times \{0, 1\}^{N \times N} \to \mathbb{R}^{N \times d}$ to produce low-dimensional node representations via raw attribute and structure contents [18], where $d$ is the dimension of the $L$-th layer, and $F$ is the dimension of the raw attribute.

**Graph neural networks.** Graph neural networks (GNNs) have been widely adopted for learning graph data by recursively aggregating and combining information from the local neighborhoods to generate expressive node- or graph-level representations [19, 20]. Each layer in a GNN follows the message-passing mechanism, always including `AGGREGATE` and `COMBINE` two stages. Given a graph $\mathcal{G}$, the output $\boldsymbol{h}_v^{(l)}$ of $l$-th GNN layer is defined as:

$$\boldsymbol{m}_v^{(l)} = \texttt{AGGREGATE}\left(\{\boldsymbol{h}_u^{(l-1)} : u \in \mathcal{N}(v)\}\right), \ \boldsymbol{h}_v^{(l)} = \texttt{COMBINE}\left(\boldsymbol{h}_v^{(l-1)}, \boldsymbol{m}_v^{(l)}\right), \quad (1)$$

where $\boldsymbol{h}_v^{(l)}$ is the representation of $v$ at $l$-th layer, and $\mathcal{N}(v)$ denotes its neighbors. The `AGGREGATE` stage gathers neighbor information and outputs aggregated information $\boldsymbol{m}_v^{(l)}$, while the `COMBINE` stage integrates it with the self-information $\boldsymbol{h}_v^{(l-1)}$. A graph-level representation is derived with `READOUT` function [21]:

$$\boldsymbol{h}_\mathcal{G} = \texttt{READOUT}\left(\{\mathbf{h}_v^{(l)} : v \in \mathcal{V}\}\right). \quad (2)$$

**Graph contrastive learning.** A typical GCL framework begins by implementing two augmentation functions $\mathcal{T}_U$ and $\mathcal{T}_V$ on the input graph $\mathcal{G} = (\boldsymbol{X}, \boldsymbol{A})$ to obtain two augmented views $U = \mathcal{T}_U(\mathcal{G})$ and $V = \mathcal{T}_V(\mathcal{G})$. These augmented graphs are processed by a GNN encoder $f(\cdot)$ to generate representations $\boldsymbol{H}_U = f(U)$ and $\boldsymbol{H}_V = f(V)$.

To train the model, a contrastive objective follows the contrastive supervision that the corresponding samples in the two views are positive, while other samples are negative. For a sample pair $(u_i, v_i)$, the contrastive loss based on the InfoNCE principle [22] is defined as:

$$\ell_{ct}(\boldsymbol{h}_{u_i}, \boldsymbol{h}_{v_i}) = -\log \frac{\exp\left(\mathcal{S}(\boldsymbol{h}_{u_i}, \boldsymbol{h}_{v_i})/\tau\right)}{\sum_j \exp\left(\mathcal{S}(\boldsymbol{h}_{u_i}, \boldsymbol{h}_{v_j})/\tau\right) + \sum_{j \neq i} \exp\left(\mathcal{S}(\boldsymbol{h}_{u_i}, \boldsymbol{h}_{u_j})/\tau\right)}, \quad (3)$$

where $\mathcal{S}(\cdot, \cdot)$ denotes a similarity function, and $\tau$ is a temperature coefficient. The overall objective sums the contrastive losses over all nodes:

$$\mathcal{L}_{ct} = \frac{1}{2N} \sum_{i=1}^N \left(\ell_{ct}(\boldsymbol{h}_{u_i}, \boldsymbol{h}_{v_i}) + \ell_{ct}(\boldsymbol{h}_{v_i}, \boldsymbol{h}_{u_i})\right). \quad (4)$$

Though this simple-yet-effective idea that maximizes the mutual information [23] is followed by considerable works [24, 25], recent studies have raised concerns regarding the rationality of contrastive supervision, particularly in how positive and negative samples are selected. NCLA [26] notices the fact that neighboring nodes are viewed as negatives is contradictory with the homophily principle, and introduces a neighbor contrastive loss to include additional positives. HomoGCL [27] finds that a part of connected nodes belonging to different classes would degrade the model performance when treating them as positives, then leverages a Gaussian mixture model to calculate the sample similarities for assigning weights to potential positives. While these methods have made important

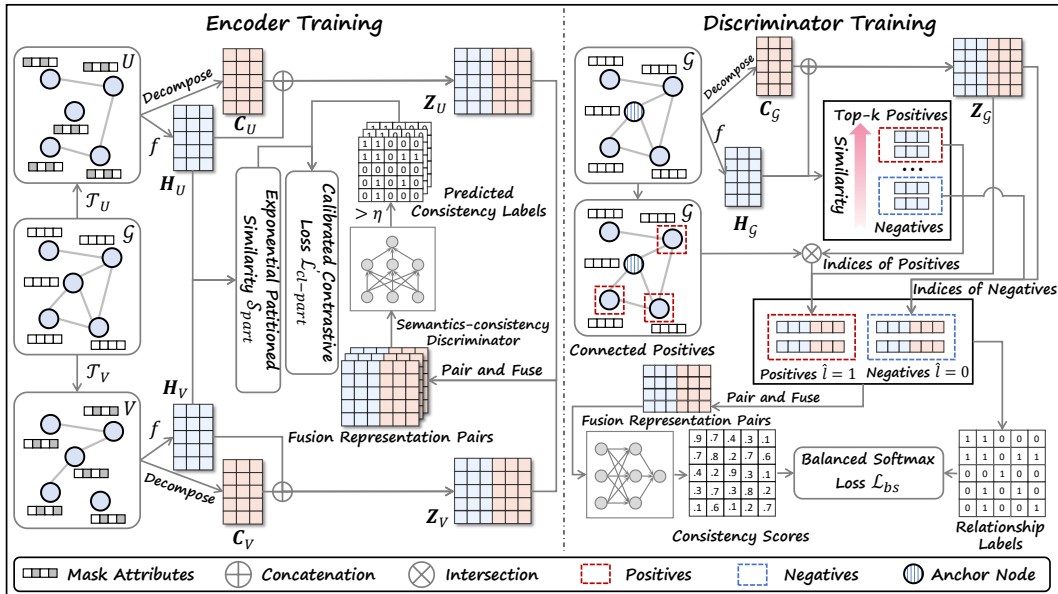

Figure 2: Training process for ClaiGCL. During encoder training, input graph $\mathcal{G}$ is transformed as views $U$ and $V$ with augmentations $\mathcal{T}_U$ and $\mathcal{T}_V$ to abstract feature information $\boldsymbol{H}_U$ and $\boldsymbol{H}_V$, and structure information $\boldsymbol{C}_U$ and $\boldsymbol{C}_V$ for constructing fusion representations $\boldsymbol{Z}_U$ and $\boldsymbol{Z}_V$. The fusion representations are constructed as inter-view and intra-view pairs that are then fused and input into the semantics-consistency discriminator to predict the relationship labels for calibrating the contrastive loss. During discriminator training, $\mathcal{G}$ is used to abstract feature information $\boldsymbol{H}_{\mathcal{G}}$ and structure information $\boldsymbol{C}_{\mathcal{G}}$ for constructing the fusion representation $\boldsymbol{Z}_{\mathcal{G}}$, where $\boldsymbol{H}_{\mathcal{G}}$ is also employed to select positives and negatives as pre-defined relationship labels with the $k$-NN algorithm and the homophily principle (i.e., connected positives). The positives and negatives are selected from $\boldsymbol{Z}_{\mathcal{G}}$ with corresponding indices, which are input into the discriminator for obtaining the consistency scores for calculating balanced softmax loss along with the relationship labels.

strides in refining contrastive supervision, most of them primarily focus on local neighborhood relationships. In contrast, our proposed CaliGCL expects to calibrate the wrong relationships from a global view, which offers a principled solution to reduce similarity estimation bias and semantic shift bias beyond local structural corrections.

## 3 CaliGCL

The training process of CaliGCL includes encoder training and discriminator training. When training the encoder, CaliGCL implements the exponential partitioned similarity measure into the contrastive loss to relieve the conflicts between positively- and negatively-synergistic features. Besides, it introduces a semantics-consistency discriminator to calibrate the semantic shift bias caused by the fixed contrastive supervision in GCL. Figure 2 shows the overview of both training processes.

### 3.1 Exponential partitioned similarity measure

When gauging the similarity of a positive pair, the positively-synergistic feature elements dominate to capture semantic consistency. However, the representation vectors are always composed of both positively- and negatively-synergistic feature elements, which easily cause conflicts in similarity calculation and obscure the semantic relationships between sample pairs.

**Theorem 1** *Given a positive representation pair $(\boldsymbol{h}_u, \boldsymbol{h}_v)$ with $d$ feature elements, where each element of $\boldsymbol{h}_u$ and $\boldsymbol{h}_v$ is independent and identically distributed (i.i.d.) following a normal distribution $\mathcal{N}(\mu, \sigma^2)$. Suppose that the expectation of the inner product of $\boldsymbol{h}_u$ and $\boldsymbol{h}_v$ is a positive value $M$, then for a partition consisting of $m$ ($m \leq d$) features, the probability that the inner product over*

*this partition is a negative value is* $\Phi\left(\frac{-m\mu^2}{\sqrt{m(\mu^2+2\sigma^2)^2}}\right)$, *where* $\Phi(\cdot)$ *is the cumulative distribution function of the standard normal distribution.*

This theorem provides the possibility that negatively-synergistic features exist in a positive pair. To address this, we introduce an exponential partitioned similarity measure, specifically designed to reduce the impairments of negatively-synergistic feature elements while enhancing the influence of positively-synergistic ones, thereby mitigating the conflicts between the two types of features and strengthening semantic alignment in representation learning.

In our design, a representation is split into $K$ fine-grained partitions, with each containing a subset of the feature elements. Thus, for the representation $\boldsymbol{h}_{v_i} = f(v_i) \in \mathbb{R}^d$ of node $v_i$, it can be represented as $\boldsymbol{h}_{v_i} = [\boldsymbol{h}_{v_i}^{(1)} \parallel \boldsymbol{h}_{v_i}^{(2)} \parallel \cdots \parallel \boldsymbol{h}_{v_i}^{(K)}]$, where $\parallel$ is the concatenation operator, and $\{\boldsymbol{h}_{v_i}^{(k)}\}_{k=1}^K$ represents the different partitions of $\boldsymbol{h}_{v_i}$. Among these partitions, each of the first $(K-1)$ partitions contains $m$ feature elements, and the last partition has $d - (K-1)m$ feature elements, where $m$ is a hyperparameter to control the partition size.

We use $\mathcal{S}_{part}(\boldsymbol{h}_{v_i}, \boldsymbol{h}_{v_j})$ to represent the exponential partitioned similarity between a sample pair $(v_i, v_j)$, which is mathematically defined as the sum of inner product similarity over $K$ partitions:

$$\mathcal{S}_{part}(\boldsymbol{h}_{v_i}, \boldsymbol{h}_{v_j}) = \frac{1}{K} \sum_{k=1}^K \exp\left(\frac{K \cdot \boldsymbol{h}_{v_i}^{(k)\top} \boldsymbol{h}_{v_j}^{(k)}}{\tau}\right), \tag{5}$$

where $\tau$ is the temperature coefficient. Unlike the commonly-used exponential inner product in contrastive learning, i.e., $\exp(\boldsymbol{h}_{v_i}^\top \boldsymbol{h}_{v_j}/\tau)$, our method computes the inner product for each partition independently before applying the exponential sum on the results. This design naturally suppresses the impact of negatively-aligned partitions by inducing their exponentials to approach zero while amplifying positively-aligned ones. For instance, given a positive pair $(v_i, v_j)$ having positive inner product is divided into two partitions where the first is negatively-aligned and the second positively-aligned, it is easy to find the following conclusion:

$$\exp\left(\frac{2 \cdot \boldsymbol{h}_{v_i}^{(1)\top} \boldsymbol{h}_{v_j}^{(1)}}{\tau}\right) \to 0, \ and \ \exp\left(\frac{2 \cdot \boldsymbol{h}_{v_i}^{(2)\top} \boldsymbol{h}_{v_j}^{(2)}}{\tau}\right) \geq \exp\left(\frac{\boldsymbol{h}_{v_i}^\top \boldsymbol{h}_{v_j}}{\tau}\right). \tag{6}$$

This holds since $\boldsymbol{h}_{v_i}^\top \boldsymbol{h}_{v_j} = \boldsymbol{h}_{v_i}^{(1)\top} \boldsymbol{h}_{v_j}^{(1)} + \boldsymbol{h}_{v_i}^{(2)\top} \boldsymbol{h}_{v_j}^{(2)}$ and $\boldsymbol{h}_{v_i}^{(2)\top} \boldsymbol{h}_{v_j}^{(2)} > \boldsymbol{h}_{v_i}^\top \boldsymbol{h}_{v_j}$ Thus, the overall partitioned similarity is more influenced by positively-synergistic components. Furthermore, we provide the following theorem to explain that the proposed exponential partitioned similarity is larger than the standard exponential inner product across all scenarios.

**Theorem 2** *The exponential partitioned similarity provides an upper bound on the standard exponential inner product similarity between two representations.*

This similarity measure will be used in the contrastive loss for debiasing the similarity conflicts.

### 3.2  Semantics-consistency discriminator

During the contrastive training, random augmentations may flip the semantic consistency between sample pairs, which implies that positive and negative sample pairs convert to each other potentially. This is difficult to perceive with the fixed contrastive supervision that the corresponding samples across different augmented views are positives, while others are negatives. To alleviate the harm from the strong assumption, we propose a semantics-consistency discriminator to dynamically evaluate the semantic relationships among sample pairs during the training process. The discrimination process includes two parts: (1) feature and structure encoding, and (2) semantics-consistency discrimination.

**Feature and structure encoding.** To provide sufficient pair information to support semantics-consistency judgements, we incorporate both feature and structural information as the input for the discriminator. For a sample pair $(v_i, v_j)$, we employ a GNN encoder to generate the corresponding representation pair $(\boldsymbol{h}_{v_i}, \boldsymbol{h}_{v_j})$ as the feature information. Meanwhile, we decompose the graph Laplacian matrix to obtain the eigenvectors as node structural information [28], which is defined as:

$$\tilde{\boldsymbol{A}} = \boldsymbol{U}\boldsymbol{\Lambda}\boldsymbol{U}^\top. \tag{7}$$

$\tilde{A}$ is the normalized adjacency matrix as [29], and it is defined as $\tilde{A} = \hat{D}^{-\frac{1}{2}} \hat{A} \hat{D}^{-\frac{1}{2}}$, where $\hat{A} = A + I_N$, $\hat{D}$ is a diagonal matrix, and its diagonal element is $\hat{D}_{ii} = \sum_j \hat{A}_{ij}$. $U$ is an orthogonal matrix constructed with eigenvectors, and $\Lambda$ is a diagonal matrix filled with eigenvalues. We select $t$ eigenvectors corresponding to top-$t$ greatest eigenvalues as the final node structure embeddings, which is denoted as $C \in \mathbb{R}^{N \times t}$, and denote the structure embeddings for $v_i$ and $v_j$ as $c_{v_i}$ and $c_{v_j}$, respectively.

**Semantics-consistency discrimination.** In this process, the feature encoding $(h_{v_i}, h_{v_j})$ and structural encoding $(c_{v_i}, c_{v_j})$ are concatenated to form a new fusion representation pair as $(z_{v_i} = h_{v_i} \parallel c_{v_i}, z_{v_j} = h_{v_j} \parallel c_{v_j})$. Then the representation pair $(z_{v_i}, z_{v_j})$ containing both the feature and structural information is used for evaluating the semantic consistency. Specifically, the representation pair $(z_{v_i}, z_{v_j})$ is first fused and then input into the semantics-consistency discriminator $D(\cdot)$ that is defined as a multi-layer perceptron (MLP) with a non-linear activation function. The process is defined as:

$$p_{v_i,v_j} = D(z_{v_i} \odot z_{v_j}), \tag{8}$$

where the Hadamard product $\odot$ is used to fuse $z_{v_i}$ and $z_{v_j}$, $p_{v_i,v_j}$ is the predicted consistency score of pair $(z_{v_i}, z_{v_j})$ by the discriminator. Finally, $p_{v_i,v_j}$ is used to judge the semantic relationship between $v_i$ and $v_j$ according to the following rule:

$$\tilde{l}_{v_i,v_j} = \begin{cases} 1, & \text{if } p_{v_i,v_j} \geq \eta \\ 0, & \text{if } p_{v_i,v_j} < \eta \end{cases}, \tag{9}$$

where $\eta$ is the threshold. $\tilde{l}_{v_i,v_j}$ is defined as the predicted consistency label, where $\tilde{l}_{v_i,v_j} = 1$ means the pair $(v_i, v_j)$ is positive, and $\tilde{l}_{v_i,v_j} = 0$ denotes the pair is negative. This label is critical for recognizing the varying relationships of sample pairs.

### 3.3 Training strategy for CaliGCL

**Training overview.** The training strategy for CaliGCL is composed of a pre-training process and a fine-tuning process. The pre-training process trains the GNN encoder to produce similarity-debiased representations by integrating the exponential partitioned similarity into the contrastive learning and develops the ability of the semantics-consistency discriminator for capturing the changing semantics of sample pairs. The fine-tuning process adopts an alternating algorithm: the discriminator is employed to calibrate the contrastive loss by correcting false positive or negative sample pair assignments caused by augmentation-induced semantic flips. Then the calibrated encoder generates more reliable representations to further enhance the discriminator in distinguishing the semantics-consistency relationships. The training strategy for improving both representation quality and semantic discrimination is detailed in the following.

#### 3.3.1 Pre-training process

**Pre-training for similarity-debiased GNN.** Following the standard GCL paradigm, we implement the augmentations on the raw graph to generate two augmented graphs $U$ and $V$, where each node $u_i \in U$ and its counterpart $v_i \in V$ form a positive pair, while others are negatives. The key difference in our framework lies in replacing the exponential inner product similarity function with the proposed exponential partitioned similarity in the contrastive loss, which is called the partitioned contrastive loss here. And the partitioned contrastive loss for $u_i$ is defined as

$$\ell_{cl-part}(u_i) = -\log \frac{\mathcal{S}_{part}(h_{u_i}, h_{v_i})}{\sum_j \mathcal{S}_{part}(h_{u_i}, h_{v_j}) + \sum_{j \neq i} \mathcal{S}_{part}(h_{u_i}, h_{u_j})}, \tag{10}$$

To ensure sufficient interactions among feature dimensions across partitions, we shuffle the feature columns of representations before loss computation. And the overall loss is given by:

$$\mathcal{L}_{cl-part} = \frac{1}{2N} \sum_{i=1}^{N} [\ell_{ct-part}(u_i) + \ell_{ct-part}(v_i)]. \tag{11}$$

**Theorem 3** *Assume the similarity of each partition for positive pairs $\mathcal{S}_{part}^+$ obeys $\mathcal{N}(\mu_1, \sigma_1^2)$, and the similarity of each partition for negative pairs $\mathcal{S}_{part}^-$ obeys $\mathcal{N}(\mu_2, \sigma_2^2)$. If $\sigma_1^2 > \sigma_2^2$ and the partition*

*similarities are i.i.d., then the partitioned contrastive objective serves as a tighter lower bound on mutual information, and yields an upper-bounded estimate compared with the contrastive objective.*

The assumption that $\sigma_1^2 \geq \sigma_2^2$ is reasonable based on the fact that the positive samples are similar because of the existence of positively-synergistic feature elements, while other negatively-synergistic feature elements would cause great divergence. In contrast, negative pairs tend to be uniformly dissimilar, leading to lower overall variance in their feature alignment. This variance divergence across partitions results in a sharper separation under exponential partitioned similarity, thus achieving a more ideal contrastive objective.

**Pre-training for semantics-consistency discriminator.** The key to training the discriminator depends on adopting reliable positive and negative pairs with pre-defined relationship labels. Specifically, a sample pair $(v_i, v_j)$ is assigned a positive relationship label $\hat{l}_{v_i, v_j} = 1$ only if the samples are both directly connected (topology proximity) and $k$-NN neighbors (feature closeness). Otherwise, we assign $\hat{l}_{v_i, v_j} = 0$ and treat the pairs as negative. Since the pre-trained GNN encoder still suffers from the semantic shift bias, the $k$-NN algorithm is applied in the original feature space but not the representation space.

Let $N_{pos}$ and $N_{neg}$ denote the total number of pre-defined positive and negative pairs, respectively. Due to $N_{pos} \ll N_{neg}$ in general, the imbalance leads the discriminator to overly favor negative predictions. Thus, we employ the Balanced Softmax loss [30, 31] to mitigate the imbalance problem:

$$\mathcal{L}_{bs} = -\sum_{i,j} \log \frac{\mathbb{I}(\hat{l}_{v_i, v_j} = 1) \exp(p_{v_i, v_j} + \log N_{pos}) + \mathbb{I}(\hat{l}_{v_i, v_j} = 0) \exp(p_{v_i, v_j} + \log N_{neg})}{\exp(p_{v_i, v_j} + \log N_{pos}) + \exp(p_{v_i, v_j} + \log N_{neg})},$$
(12)

where $p_{v_i, v_j}$ is the predicted consistency score for the pair $(v_i, v_j)$ according to Eq. (8), and $\mathbb{I}(\cdot)$ is the indicator function. This loss effectively prevents the prediction bias and enables the discriminator to more accurately distinguish semantically consistent pairs.

### 3.3.2 Fine-tuning process

After the pre-training process, the initially calibrated graph encoder could produce relatively high-quality representations, while the relationship-consistency discriminator possesses a certain ability to detect semantic changes among sample pairs. It is proper to further enhance both components and drive them towards mutual reinforcement. Considering this, we adopt an alternating training strategy between the GNN encoder and the discriminator.

**Encoder fine-tuning with calibrated supervision.** In each iteration, we first obtain two augmented graph views $U$ and $V$ of the input graph with graph augmentations, then follow the procedure in Section 3.2 to yield the fusion representation matrices $\boldsymbol{Z}_U$ and $\boldsymbol{Z}_V$ for the two augmented views. These matrices are used for constructing the intra-view pairs $(\boldsymbol{Z}_U, \boldsymbol{Z}_U)$, $(\boldsymbol{Z}_V, \boldsymbol{Z}_V)$ and inter-view pair $(\boldsymbol{Z}_U, \boldsymbol{Z}_V)$ as the input of the discriminator. The discriminator then estimates the consistency score of each sample pair among these fusion representation pairs to further calibrate the encoder through the calibrated contrastive loss:

$$\ell_{cali}(u_i) =$$
$$-\log \frac{\sum_j \tilde{l}_{u_i, v_j} \cdot \mathcal{S}_{part}(\boldsymbol{h}_{u_i}, \boldsymbol{h}_{v_j}) + \sum_{j \neq i} \tilde{l}_{u_i, u_j} \cdot \mathcal{S}_{part}(\boldsymbol{h}_{u_i}, \boldsymbol{h}_{u_j})}{\sum_i \tilde{l}_{u_i, v_i} \cdot \mathcal{S}_{part}(\boldsymbol{h}_{u_i}, \boldsymbol{h}_{v_i}) + \sum_j \tilde{l}'_{u_i, v_j} \cdot \mathcal{S}_{part}(\boldsymbol{h}_{u_i}, \boldsymbol{h}_{v_j}) + \sum_{j \neq i} \tilde{l}'_{u_i, u_j} \cdot \mathcal{S}_{part}(\boldsymbol{h}_{u_i}, \boldsymbol{h}_{u_j})},$$
(13)

where $\tilde{l}_{u_i, v_j} = \mathbb{I}(p_{u_i, v_j} \geq \eta)$ represents the consistency label between $u_i$ and $v_j$ by inputting the corresponding fusion representation into the discriminator, and $\tilde{l}'_{u_i, v_j} = 1 - \tilde{l}_{u_i, v_j}$. Additionally, we only add the positive sample pairs with the highest confidence in the denominator to reduce computation cost, where an adopted sample pair is judged as positive by the discriminator and includes corresponding samples between the augmentation views. This calibrated contrastive loss promotes alignment between consistent pairs while suppressing misleading or semantically flipped pairs that arise from graph augmentations. Similarly, we have the overall loss for fine-tuning GNN as

$$\mathcal{L}_{cali} = \frac{1}{2N} \sum_{i=1}^{N} [\ell_{cali}(u_i) + \ell_{cali}(v_i)].$$
(14)

Table 1: Empirical results of self-supervised representation learning in node classification. The best results are highlighted in **bold**, and the second-best results are highlighted with underline.

| Dataset | Cora | Citeseer | Pubmed | DBLP | Photo | Computers |
|---------|------|----------|--------|------|-------|-----------|
| AFGRL | 83.34 ± 0.87 | 71.49 ± 0.78 | 85.21 ± 0.22 | 83.08 ± 0.14 | 93.22 ± 0.28 | 89.88 ± 0.33 |
| SUGRL | 85.37 ± 0.55 | 73.49 ± 0.66 | 86.68 ± 0.21 | 83.03 ± 0.23 | 93.20 ± 0.40 | 88.90 ± 0.20 |
| NCLA | 85.32 ± 0.44 | 73.24 ± 0.55 | 85.47 ± 0.40 | 83.97 ± 0.17 | 93.48 ± 0.22 | 89.14 ± 0.35 |
| GREET | 85.70 ± 0.46 | 73.26 ± 0.60 | 86.95 ± 0.31 | 83.81 ± 0.13 | 92.85 ± 0.31 | 87.94 ± 0.35 |
| HomoGCL | 85.40 ± 0.46 | 72.34 ± 0.40 | 86.31 ± 0.18 | 84.39 ± 0.16 | 93.25 ± 0.26 | 90.09 ± 0.32 |
| S2GAE | 85.46 ± 0.28 | 73.37 ± 0.47 | 86.46 ± 0.09 | 84.08 ± 0.00 | 93.50 ± 0.16 | 89.99 ± 0.14 |
| iGCL | 84.31 ± 0.43 | 72.85 ± 0.85 | 86.17 ± 0.25 | 83.74 ± 0.23 | 93.10 ± 0.26 | 90.06 ± 0.41 |
| PiGCL | 84.63 ± 0.78 | 73.51 ± 0.64 | 86.30 ± 0.20 | 84.30 ± 0.28 | 93.14 ± 0.30 | 89.25 ± 0.27 |
| Bandana | 85.48 ± 0.84 | 73.04 ± 0.75 | 86.35 ± 0.25 | 83.93 ± 0.02 | 93.44 ± 0.11 | 89.62 ± 0.09 |
| SGRL | 85.36 ± 0.23 | 73.20 ± 0.22 | 86.17 ± 0.01 | 84.12 ± 0.01 | **93.83 ± 0.04** | 90.02 ± 0.02 |
| CaliGCL | **85.87 ± 0.62** | **74.13 ± 0.54** | **87.16 ± 0.19** | **85.32 ± 0.13** | 93.72 ± 0.24 | **90.79 ± 0.28** |

**Discriminator enhancement with updated representations.** Discriminator fine-tuning is similar to the pre-training, but differs in the sample selection strategy. Since the semantic shift bias of the encoder is alleviated by the discriminator, the produced representations of the encoder are more reliable. Considering this, we apply the $k$-NN algorithm in the representation space to pick out the positives and negatives rather than the original feature space. These pairs are used to update the discriminator with Eq. (12), allowing it to more accurately model semantic consistency based on progressively refined representations.

## 4 Experiments

In this section, we evaluate our proposed model across multiple widely used graph datasets, including citation networks (**Cora**, **Citeseer**, **Pubmed**, and **DBLP**), co-purchase networks (**Amazon Photo**, and **Amazon Computers**), social networks (**COLLAB**, **REDDIT-BINARY**, **REDDIT-MULTI-5K**, **IMDB-BINARY**), and biochemical networks (**NCI1**, **PROTEINS**, **DD**, and **MUTAG**). The detailed dataset information and hyperparameter settings for these datasets are provided in the Appendix B. In the comparative experiments, the node classification and graph classification are employed to evaluate the model expressiveness. For fair comparison, we compare our model with recent state-of-the-art self-supervised graph models for each task. All the experiments are implemented in Pytorch and conducted on a server with RTX 3090 (24 GB).

### 4.1 Node classification

In node classification, we compare our model against both contrastive and generative graph representation learning methods, including **AFGRL** [11], **SUGRL** [32], **NCLA** [26], **GREET** [33], **HomoGCL** [27], **S2GAE** [34], **iGCL** [35], **PiGCL** [36], **Bandana** [37], and **SGRL** [38]. In our evaluation pipeline, the raw graph is first encoded to produce node representations, which are then employed for training a linear classifier. Due to the weakness of the linear classifier, the performance mainly depends on the representation quality. In our experiment setting, we split 10% representations for training the classifier, 10% for validation, and the remainder for testing. We report the mean F1 with standard deviation to evaluate the performance, and repeat experiments 10 times to provide reliable outcomes in Table 1.

The comparison results reveal that our model is competitive or even surpasses other baselines, which validates the model effectiveness. Additionally, we have the following observations:
(1) GCL models that solely rely on cross-view positives, such as GCA and PiGCL, show inferior performance to other models incorporating the extra neighbor positives, e.g., NCLA, GREET, and HomoGCL. It suggests the significance of calibrating the contrastive supervision.
(2) Generative models like S2GAE and Bandana, which focus on reconstructing graph topology, achieve performance on par with powerful contrastive methods. This supports the homophily principle and further explains why those GCL models focusing on local consistency perform well.
(3) The proposed CaliGCL effectively calibrates the similarity estimation bias and semantic shift bias in GCL, showing best performance on nearly all the datasets. It signifies that CaliGCL has sufficient

Table 2: Comparisons with different graph models in graph classification. The best results are highlighted in **bold**, and the second-best results are highlighted with underline.

| Datasets | Biochemical Molecules | | | Social Networks | | | |
|---|---|---|---|---|---|---|---|
| | NCI1 | DD | MUTAG | COLLAB | RDT-B | RDT-M5K | IMDB-B |
| InfoGCL | 76.2 ± 1.1 | 72.9 ± 1.8 | 89.0 ± 1.1 | 70.7 ± 1.1 | 82.5 ± 1.4 | 53.5 ± 1.0 | 73.0 ± 0.9 |
| GraphCL | 77.9 ± 0.4 | 78.6 ± 0.4 | 86.8 ± 1.4 | 71.4 ± 1.2 | 89.5 ± 0.8 | 56.0 ± 0.3 | 71.1 ± 0.4 |
| JOAO | 78.1 ± 0.5 | 77.3 ± 0.5 | 87.4 ± 1.0 | 69.5 ± 0.4 | 85.3 ± 1.4 | 55.7 ± 0.6 | 70.2 ± 3.1 |
| JOAOv2 | 78.4 ± 0.5 | 77.4 ± 1.2 | 87.7 ± 0.8 | 69.3 ± 0.3 | 86.4 ± 1.5 | 56.0 ± 0.2 | 70.8 ± 0.3 |
| AD-GCL | 69.7 ± 0.5 | 74.5 ± 0.5 | 88.6 ± 1.3 | 73.3 ± 0.6 | 85.5 ± 0.8 | 53.0 ± 0.8 | 71.6 ± 1.0 |
| SimGRACE | 79.1 ± 0.4 | 77.4 ± 1.1 | 89.0 ± 1.3 | 71.7 ± 0.8 | 89.5 ± 0.9 | 55.9 ± 0.3 | 71.3 ± 0.8 |
| SPAN | 71.4 ± 0.5 | 75.8 ± 0.5 | 89.1 ± 0.8 | 75.0 ± 0.5 | 83.6 ± 0.6 | 54.1 ± 0.5 | 73.7 ± 0.7 |
| GPA | 80.4 ± 0.4 | 79.9 ± 0.4 | 89.7 ± 0.8 | 76.2 ± 0.1 | 89.3 ± 0.4 | 53.7 ± 0.2 | 74.6 ± 0.4 |
| DRGCL | 79.7 ± 0.4 | 78.4 ± 0.7 | 89.5 ± 0.6 | 70.6 ± 0.8 | **90.8 ± 0.3** | 56.3 ± 0.2 | 72.0 ± 0.5 |
| CaliGCL | **80.8 ± 2.0** | **80.4 ± 2.5** | **90.4 ± 5.8** | **77.2 ± 1.7** | 90.3 ± 2.7 | **56.5 ± 1.5** | **76.9 ± 5.0** |

competence to distinguish its positives and negatives, benefiting the downstream classification task. (4) While CaliGCL performs well on smaller datasets such as Cora and Citeseer, it exhibits even stronger improvements on larger datasets like DBLP and Computers. This highlights its ability to uncover and utilize latent positives that are often neglected by the conventional contrastive loss.

The superior performance of CaliGCL stems from its ability to calibrate both the similarity estimation bias and semantic shift bias in GCL, which allows CaliGCL to identify latent positive pairs and remove spurious negatives, ultimately leading to more discriminative representations.

## 4.2 Graph classification

In this experiment, we evaluate the performance of CaliGCL on the graph classification task under an unsupervised learning setting, following standard protocols introduced in [39] and [40]. Different from the node classification, the positives based on the $k$-NN algorithm are difficult to obtain since graph datasets may lack feature information. Thus, the discriminator is not pre-trained and ignores the structure information.

For the experiment setting, we first train a GIN encoder and then train an SVM classifier to classify the produced graph representations via 10-fold cross-validation. Our proposed model is compared with recent self-supervised graph models, including **InfoGCL** [41], **GraphCL** [39], **JOAO** and **JOAOv2** [42], **AD-GCL** [43], **SimGRACE** [43], **SPAN** [44], **GPA** [40], and **DRGCL** [45]. According to the results in Table 2, we have the following observations:
(1) When compared with the augmentation-free model SimGRACE, CaliGCL preserves the augmentations to realize sample diversity, which is crucial for effective GCL learning. Moreover, CaliGCL mitigates the adverse effects of the augmentations by calibrating the biases inherent in GCL.
(2) CaliGCL still outperforms models that employ learnable augmentations, such as GPA. This superiority is attributed to the competence of debiasing semantic shift, which is also neglected by models that only optimize augmentation strength or combination strategies, without considering their potential to distort semantic consistency between sample pairs.

Overall, the graph classification results confirm that CaliGCL could also boost the graph-level task by adapting a contrastive learning framework to work on graph-level representations. This highlights the flexibility and robustness of the proposed framework in self-supervised learning settings.

## 4.3 Hyperparameter analysis

we conduct a sensitivity analysis on the temperature coefficient $\tau$ and the partition size $m$, and report the mean F1 as the model performance in Figure 3. Figure 3(a) and (b) show that the performance changes over different values of $\tau$, indicating that the temperature coefficient is critical for distinguishing the positives and negatives. Figure 3(c) and (d) demonstrate that partitions with an appropriate number of feature elements $m$ can improve the model performance. This supports the effectiveness of partitioning the representation vectors to calibrate fine-grained semantic structures. However, more partitions may introduce variance and degrade the performance.

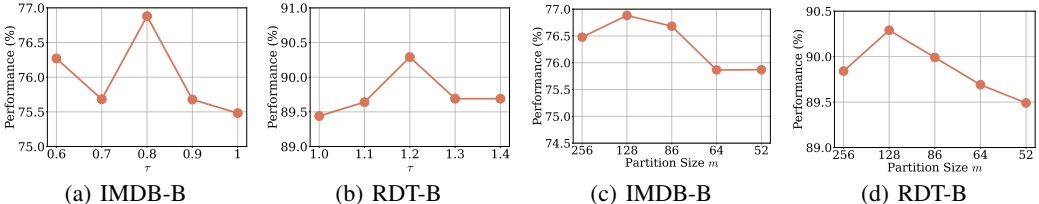

| (a) IMDB-B | (b) RDT-B | (c) IMDB-B | (d) RDT-B |

Figure 3: Hyperparameter analysis on the temperature coefficient $\tau$ and the partition size $m$.

## 5 Conclusion

In this paper, we identify two biases in GCL: similarity estimation bias caused by conflicts between positively- and negatively-synergistic features, and semantic shift bias introduced by random augmentations. To mitigate the issues, we propose CaliGCL to enhance the influence of positively-synergistic features while suppressing negative ones, and correct semantic inconsistencies in contrastive pairs. However, the exponential partitioned similarity measure is not adaptive, and it could be further enhanced by automatically distinguishing the positively- and negatively-synergestic features.

## Acknowledgments and Disclosure of Funding

This work was supported by the National Natural Science Foundation of China (No. 62306020), the Young Elite Scientist Sponsorship Program by BAST (No. BYESS2024199), Beijing Natural Science Foundation (No. L244009), the National Key Research and Development Program of China (No. 2023YFB3107100), and partially supported by the Central Guidance for Local Scientific and Technological Development Fund (No.2024ZY0124). The first author is funded by the China Scholarship Council (CSC) from the Ministry of Education, China.

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

## A Theorem proofs

### A.1 The proof of Theorem 1

**Proof.** Let $h_{u,i}$ and $h_{v,i}$ be the $i$-th elements of $h_u$ and $h_v$, respectively. Since $h_{u,i}$ and $h_{v,i}$ are i.i.d., then the expectation of $h_{u,i}h_{v,i}$ is

$$\mathbb{E}[h_{u,i}h_{v,i}] = \mathbb{E}[h_{u,i}]\mathbb{E}[h_{v,i}] = \mu^2. \tag{15}$$

Based on this, the expectation of the inner product over $m$ ($m \leq d$) features is

$$\sum_{i=1}^{m} \mathbb{E}[h_{u,i}h_{v,i}] = m\mu^2. \tag{16}$$

For $h_{u,i}, h_{u,i} \sim \mathcal{N}(\mu, \sigma^2)$, we have

$$\mathbb{E}[h_{u,i}^2] = \text{Var}(h_{u,i}) + (\mathbb{E}[h_{u,i}])^2 = \sigma^2 + \mu^2, \tag{17}$$

Similarly, we have $\mathbb{E}[h_{v,i}^2] = \sigma^2 + \mu^2$. Thus, the expectation of $h_{u,i}^2 h_{v,i}^2$ is

$$\mathbb{E}[h_{u,i}^2 h_{v,i}^2] = (\sigma^2 + \mu^2)^2. \tag{18}$$

Then the variance of $h_{u,i}h_{v,i}$ is calculated as

$$\text{Var}(h_{u,i}h_{v,i}) = \mathbb{E}[h_{u,i}^2 h_{v,i}^2] - (\mathbb{E}[h_{u,i}h_{v,i}])^2 = (\sigma^2 + \mu^2)^2 - \mu^4 = \mu^4 + 2\mu^2\sigma^2. \tag{19}$$

Since $h_{u,i}$ and $h_{v,i}$ are independent, we have

$$\text{Var}\left(\sum_{i=1}^{m} h_{u,i}h_{v,i}\right) = m(\mu^4 + 2\mu^2\sigma^2). \tag{20}$$

Therefore, by the Central Limit Theorem, we have $\sum_{i=1}^{m} h_{u,i}h_{v,i} \sim \mathcal{N}(m\mu^2, m(\mu^2 + 2\sigma^2)^2)$. Finally, the probability that $\sum_{i=1}^{m} h_{u,i}h_{v,i} < 0$ is

$$\mathbb{P}\left(\sum_{i=1}^{m} h_{u,i}h_{v,i} < 0\right) = \Phi\left(\frac{-m\mu^2}{\sqrt{m(\mu^2 + 2\sigma^2)^2}}\right). \tag{21}$$

### A.2 The proof of Theorem 2

**Proof.** We consider the temperature coefficient $\tau = 1$ for both the exponential partitioned similarity and the exponential inner product similarity. For a pair $(h_u, h_v)$, we have the standard exponential inner product similarity defined as:

$$\mathcal{S}(h_u, h_v) = \exp(h_u^\top h_v) = \exp\left(\sum_{k=1}^{K} h_u^{(k)\top} h_v^{(k)}\right),$$

where $\mathcal{S}(\cdot)$ denotes the exponential inner product similarity, and $h_u^{(k)}$, $h_v^{(k)}$ are the $k$-th partitions of representations $h_u$ and $h_v$, respectively. The exponential partitioned similarity measure is defined as

$$\mathcal{S}_{part}(h_u, h_v) = \frac{1}{K}\sum_{k=1}^{K} \exp\left(h_u^{(k)\top} h_v^{(k)}\right). \tag{22}$$

By Jensen's inequality and the convexity of the exponential function, for all $\mathcal{S}_1, \mathcal{S}_2, \ldots, \mathcal{S}_K \in \mathbb{R}$ (each representing the inner product of a partition), we have

$$\exp\left(\frac{1}{K}\sum_{k=1}^{K} \mathcal{S}_k\right) \leq \frac{1}{K}\sum_{k=1}^{K} \exp(\mathcal{S}_k), \tag{23}$$

which leads to

$$\exp\left(\sum_{k=1}^{K} \mathcal{S}_k\right) \leq \frac{1}{K}\sum_{k=1}^{K} \exp(K\mathcal{S}_k), \tag{24}$$

This inequality corresponds to

$$\mathcal{S}(h_u, h_v) \leq \mathcal{S}_{part}(h_u, h_v). \tag{25}$$

## A.3 The proof of Theorem 3

To prove the theorem, we first provide the following **Lemma 4** as

**Lemma 4** *Let $\boldsymbol{h}$ and $\boldsymbol{h}^+$ be positive sample pairs drawn from a joint distribution $p(\boldsymbol{h}, \boldsymbol{h}^+)$, and let $\mathcal{S}_{part}(\boldsymbol{h}, \boldsymbol{h}') = \frac{1}{K} \sum_{k=1}^{K} \exp\left(\frac{K}{\tau} \cdot \boldsymbol{h}^{(k)\top} \boldsymbol{h}'^{(k)}\right)$ denote the exponential partitioned similarity between two representations. Define the partitioned contrastive loss as:*

$$\mathcal{L}_{part} = -\mathbb{E}_{\boldsymbol{h}, \boldsymbol{h}^+} \left[ \log \frac{\mathcal{S}_{part}(\boldsymbol{h}, \boldsymbol{h}^+)}{\sum_{\boldsymbol{h}'} \mathcal{S}_{part}(\boldsymbol{h}, \boldsymbol{h}')} \right],$$

*Then, the following lower bound on mutual information holds:*

$$I(\boldsymbol{h}; \boldsymbol{h}^+) \geq \log N - \mathcal{L}_{part}.$$

*where $N$ is the number of samples.*

**Proof.** This result follows from the lower bound of mutual information derived in [22], which shows that any InfoNCE-style loss of the form:

$$\mathcal{L}_{NCE} = -\mathbb{E}_{\boldsymbol{h}, \boldsymbol{h}^+} \left[ \log \frac{p(\boldsymbol{h}^+ | \boldsymbol{h})}{\sum_{\boldsymbol{h}'} p(\boldsymbol{h}' | \boldsymbol{h})} \right] \tag{26}$$

satisfies the lower bound $I(\boldsymbol{h}; \boldsymbol{h}^+) \geq \log N - \mathcal{L}_{NCE}$ for any positive scoring function $p(\boldsymbol{h}^+ | \boldsymbol{h})$ approximating the true density ratio. In our case, we define $p(\boldsymbol{h}^+ | \boldsymbol{h}) \propto \mathcal{S}_{part}(\boldsymbol{h}, \boldsymbol{h}^+)$, which is strictly positive and normalized over the dataset set via softmax. Therefore, the loss $\mathcal{L}_{part}$ conforms to the InfoNCE formulation and inherits the same mutual information lower bound.

Based on **Lemma 4**, the proof of **Theorem 3** is transformed to prove that $\mathcal{L}_{part}$ is the lower bound of $\mathcal{L}_{NCE}$. Therefore, we provide the following proof.

For a positive pair, its inner product similarity $\mathcal{S}^+$ obeys the following distribution:

$$\mathcal{S}^+ = \sum_{k=1}^{K} \mathcal{S}^{(k)+} \sim \mathcal{N}(K\mu_1, K\sigma_1^2). \tag{27}$$

Similarly, the distribution for the inner product similarity of a negative pair $\mathcal{S}^-$ is

$$\mathcal{S}^- = \sum_{k=1}^{K} \mathcal{S}^{(k)-} \sim \mathcal{N}(K\mu_2, K\sigma_2^2). \tag{28}$$

Then the original contrastive loss for a sample pair is defined as:

$$\mathcal{L}_{ct} = -\log \frac{\exp(\mathcal{S}^+)}{\exp(\mathcal{S}^+) + \sum_j \exp(\mathcal{S}_j^-)}, \tag{29}$$

where $\mathcal{S}^+$ is the inner product similarity between a positive pair, and $\mathcal{S}_j^-$ represents the inner product similarity between any other negative pair. Similarly, the partition contrastive loss is given by

$$
\begin{aligned}
\mathcal{L}_{ct-part} &= -\log \frac{\frac{1}{K} \sum_k \exp(K\mathcal{S}_{part}^{(k)+})}{\frac{1}{K} \sum_k \exp(K\mathcal{S}_{part}^{(k)+}) + \frac{1}{K} \sum_j \sum_k \exp(K\mathcal{S}_{part,j}^{(k)-})} \\
&= -\log \frac{\sum_k \exp(K\mathcal{S}_{part}^{(k)+})}{\sum_k \exp(K\mathcal{S}_{part}^{(k)+}) + \sum_j \sum_k \exp(K\mathcal{S}_{part,j}^{(k)-})}
\end{aligned} \tag{30}
$$

where $S_{part}^{(k)+}$ is the inner product similarity of the $k$-th partition between a positive pair, and $S_{part,j}^{(k)-}$ represents the inner product similarity of the $k$-th partition between a negative pair.

Let

$$
\begin{cases}
A = \exp(\mathcal{S}^+) \\
\widetilde{A} = \sum_k \exp(K\mathcal{S}_{part}^{(k)+}) \\
B = \exp(\mathcal{S}^+) + \sum_j \mathcal{S}_j^- \\
\widetilde{B} = \sum_k \exp(K\mathcal{S}_{part}^{(k)+}) + \sum_j \sum_k \exp(K\mathcal{S}_{part,j}^{(k)-})
\end{cases}, \tag{31}
$$

the comparison between the two losses is transformed to compare $\frac{\tilde{A}}{\tilde{B}}$ with $\frac{A}{B}$.

Let

$$
\begin{cases}
\alpha = \frac{\tilde{A}}{\tilde{B}} \\
\beta = \frac{A}{B} \\
P_1 = \mathbb{E}[\exp(\mathcal{S}^+)] \\
P_2 = \mathbb{E}[\exp(K \cdot \mathcal{S}^{(k)+})] \\
Q_1 = \mathbb{E}[\exp(\mathcal{S}^-)] \\
Q_2 = \mathbb{E}[\exp(K \cdot \mathcal{S}_{part,j}^{(k)-})]
\end{cases}
, \tag{32}
$$

and we implement the expectations to approximate $\alpha$ and $\beta$:

Then we have

$$
\alpha \approx \frac{P_2}{P_2 + (N-1)Q_2}, \quad \beta \approx \frac{P_1}{P_1 + (N-1)Q_1}. \tag{33}
$$

Therefore, $\frac{\alpha}{\beta} \approx \frac{P_2[P_1+(N-1)Q_1]}{P_1[P_2+(N-1)Q_2]} = \frac{P_1P_2+(N-1)Q_1P_2}{P_1P_2+(N-1)Q_2P_1}$, it means that whether the ratio between $\alpha$ and $\beta$ is greater depends on the comparison between $Q_1P_2$ and $Q_2P_1$.

According to the assumption that the inner product similarities of partitions follow the Gaussian distribution, the exponentials of the inner product similarities obey the log-normal distribution, and we have

$$
\begin{cases}
P_1 = \mathbb{E}[\exp(\mathcal{S}^+)] = \exp(K\mu_1 + \frac{K\sigma_1^2}{2}) \\
P_2 = \mathbb{E}[\exp(K \cdot \mathcal{S}^{(k)+})] = \exp(K\mu_1 + \frac{K^2\sigma_1^2}{2}) \\
Q_1 = \mathbb{E}[\exp(\mathcal{S}^-)] = \exp(K\mu_2 + \frac{K\sigma_2^2}{2}) \\
Q_2 = \mathbb{E}[\exp(K \cdot \mathcal{S}_{part,j}^{(k)-})] = \exp(K\mu_2 + \frac{K^2\sigma_2^2}{2})
\end{cases}
. \tag{34}
$$

We turn the comparison between $Q_1P_2$ and $Q_2P_1$ to the comparison between $\log \frac{P_1}{Q_1}$ and $\log \frac{P_2}{Q_2}$, then we have

$$
\log \frac{P_1}{Q_1} - \log \frac{P_2}{Q_2} = \frac{K - K^2}{2}(\sigma_1^2 - \sigma_2^2). \tag{35}
$$

Since $K \geq 1$, $K - K^2 \leq 0$. According to the assumption $\sigma_1^2 - \sigma_2^2 \geq 0$, we have

$$
\log \frac{P_1}{Q_1} - \log \frac{P_2}{Q_2} \leq 0. \tag{36}
$$

This means that the partition contrastive loss is the lower bound of the original contrastive loss, and the partition contrastive objective is an upper bound of mutual information than the original contrastive objective.

## B   Experiment

### B.1   Datasets

We supplement the concrete descriptions of datasets in the following. Besides, the statistics of the datasets for learning node-level representations are shown in Table 3, and those for learning graph-level representations are shown in Table 4.

**Cora**, **CiteSeer**, **Pubmed**, and **DBLP** are four most widely used citation network datasets. In these datasets, each node represents a scientific paper, and the node attributes correspond to specific keywords extracted from the paper. An edge between two nodes indicates a citation relationship.

**Amazon Photo** and **Amazon Computers** are co-purchase networks, where nodes represent products and edges indicate that two products are frequently bought together. Node features are derived using a bag-of-words representation based on product descriptions.

**NCI1** is a molecular graph dataset used for predicting chemical compounds' activity against cancer cells. **MUTAG** is another molecular dataset, commonly used for mutagenicity prediction. **DD** also contains chemical compounds and is employed in general graph classification tasks.

**COLLAB** is a social network dataset used for collaborative filtering. **REDDIT-BINARY** (RDT-B) consists of Reddit threads and is used for binary community classification. **IMDB-BINARY** (IMDB-B) is a dataset derived from movie collaboration networks, labeled for binary classification. **REDDIT-MULTI5K** (RDT-M5K) is a multi-class variant of the Reddit dataset, containing threads labeled into five categories.

Table 3: Statistics of different graph datasets for learning node-level representations.

| Dataset | Cora | Citeseer | Pubmed | DBLP | Amazon Photo | Amazon Computers |
|---|---|---|---|---|---|---|
| Nodes | 2,708 | 3,327 | 19,717 | 17,716 | 7,650 | 13,752 |
| Edges | 5,429 | 4,732 | 44,338 | 52,867 | 119,081 | 491,722 |
| Features | 1,433 | 3,703 | 500 | 1,639 | 745 | 767 |
| Classes | 7 | 6 | 3 | 4 | 8 | 10 |

Table 4: Statistics of different graph datasets for learning graph-level representations.

| Data Type | Name | Graphs | Average Nodes | Average Edges | Classes |
|---|---|---|---|---|---|
| Biochemical Molecules | NCI1 | 4,110 | 29.87 | 32.30 | 2 |
| | MUTAG | 188 | 17.93 | 19.79 | 2 |
| | DD | 1,178 | 284.32 | 715.66 | 2 |
| Social Networks | COLLAB | 5,000 | 74.5 | 2457.78 | 3 |
| | REDDIT-BINARY | 2,000 | 429.6 | 497.75 | 2 |
| | REDDIT-MULTI-5K | 4,999 | 508.8 | 594.87 | 5 |
| | IMDB-BINARY | 1,000 | 19.8 | 96.53 | 2 |

## B.2 Hyperparameter Setting

The hyperparameter settings for node and graph classification are listed in Table 5 and 6, respectively. For the hyperparameters used in the node-level datasets, Lr_Enc is the learning rate of the graph encoder in the pre-training phase, Epc_Init_Enc is the training epochs of the graph encoder in the pre-training phase, and Hid_dim is the hidden dimension of the graph encoder. Similarly, Lr_Dis, Epc_Init_Dis, Dis_dim, Dis_act are the learning rate, training epochs, hidden dimension, and activation function for the discriminator in the pre-training phase, respectively. Leaky represents LeakyReLU activation function. Itr_num is the number of iterations in the alternating training strategy, where the graph encoder is trained with Epc_FT_Enc epochs in each iteration and the discriminator is trained with Epc_FT_Dis epochs in each iteration. Proj_dim is the hidden dimension of the projector head, $t$ is the dimension of the structure embedding, $m$ is the number of feature elements in a partition, $\eta$ is the threshold to distinguish the positive pairs and negative pairs for the discriminator, and $\tau$ is the temperature coefficient. $p_{e1}$, $p_{f1}$, $p_{e2}$, and $p_{f2}$ are four hyperparameters for controlling the strength of graph augmentations. The hyperparameter settings in graph-level datasets are similar to he node-level datasets, but we abandon the structure embeddings since it is not appropriate to use node structure embeddings for constructing graph structure embeddings. Augmentation is the graph-level augmentation strategy adopted by the datasets, where 'dnodes' is to drop nodes, and 'random' includes node dropping, edge perturbation, attribute masking, and subgraph sampling. All the augmentations adopt the default ratio as [39].

## B.3 Ablation study

We perform ablation studies to assess the contributions of the exponential partitioned similarity ("$\mathcal{S}_{part}$") and the semantics-consistency discriminator ("D") in the CaliGCL framework, and the results are reported in Figure 4. From the results, we could find that the exponential partitioned similarity and the discriminator could enhance the model performance. However, joint implementation of both methods can effectively alleviate the biases in GCL and further improve the model performance.

Table 5: Hyperparameter setting for node-level datasets.

| Datasets | Cora | Citeseer | Pubmed | DBLP | Photo | Computers |
|---|---|---|---|---|---|---|
| Lr_Enc | 1e-4 | 1e-3 | 3e-4 | 2e-4 | 5e-4 | 1e-3 |
| Epc_Init_Enc | 300 | 300 | 8000 | 1000 | 6000 | 9000 |
| Hid_dim | [512,256] | [512,256] | [512,256] | [512,256] | [512,256] | [512,256] |
| Lr_Dis | 4e-4 | 8e-4 | 8e-3 | 8e-3 | 8e-3 | 8e-4 |
| Dis_dim | 128 | 512 | 256 | 128 | 128 | 512 |
| Dis_act | Leaky | Leaky | Leaky | Leaky | Leaky | Leaky |
| Epc_FT_Dis | 10 | 10 | 10 | 5 | 5 | 20 |
| Itr_num | 25 | 15 | 30 | 5 | 1 | 15 |
| Epc_FT_Enc | 15 | 15 | 15 | 15 | 5 | 15 |
| Epc_Init_Dis | 25 | 10 | 25 | 150 | 40 | 25 |
| Proj_dim | 128 | 256 | 512 | 512 | 512 | 512 |
| $t$ | 64 | 64 | 64 | 64 | 64 | 64 |
| $m$ | 128 | 240 | 240 | 128 | 192 | 200 |
| $\eta$ | 0.6 | 0.6 | 0.6 | 0.6 | 0.6 | 0.6 |
| $\tau$ | 0.8 | 0.9 | 0.23 | 1.2 | 0.1 | 0.1 |
| $p_{e1}$ | 0.2 | 0.2 | 0.4 | 0.1 | 0.3 | 0.5 |
| $p_{f1}$ | 0.3 | 0.3 | 0.0 | 0.4 | 0.1 | 0.2 |
| $p_{e2}$ | 0.4 | 0.4 | 0.1 | 0.1 | 0.5 | 0.5 |
| $p_{f2}$ | 0.4 | 0.5 | 0.2 | 0.0 | 0.1 | 0.1 |

Table 6: Hyperparameter setting for graph-level datasets.

| Datasets | MUTAG | IMDB-B | COLLAB | RDT-B | DD | NCI1 | RDT-M5K |
|---|---|---|---|---|---|---|---|
| Lr_Enc | 2e-3 | 2e-3 | 1e-3 | 3e-3 | 4e-3 | 3e-3 | 1e-3 |
| Epc_Init_Enc | 40 | 40 | 270 | 80 | 90 | 270 | 270 |
| Hid_dim | [128,128] | [128,128] | [256,128] | [128,128] | [256,256] | [128,128] | [128,128] |
| Epc_FT_Enc | 15 | 15 | 15 | 15 | 15 | 15 | 15 |
| Lr_Dis | 8e-3 | 8e-3 | 8e-3 | 8e-3 | 8e-3 | 8e-3 | 8e-3 |
| Dis_dim | 128 | 128 | 128 | 128 | 64 | 64 | 128 |
| Dis_act | Leaky | Leaky | Leaky | Leaky | Leaky | Leaky | Leaky |
| Epc_FT_Dis | 5 | 5 | 5 | 5 | 10 | 10 | 5 |
| Itr_num | 5 | 5 | 5 | 5 | 5 | 5 | 5 |
| Proj_dim | 128 | 128 | 256 | 128 | 256 | 256 | 256 |
| Augmentation | dnodes | random | dnodes | dnodes | dnodes | dnodes | dnodes |
| $m$ | 128 | 128 | 128 | 128 | 128 | 256 | 128 |
| $\eta$ | 0.6 | 0.6 | 0.6 | 0.6 | 0.6 | 0.6 | 0.6 |
| $\tau$ | 0.7 | 0.8 | 0.05 | 1.2 | 0.1 | 0.05 | 0.04 |

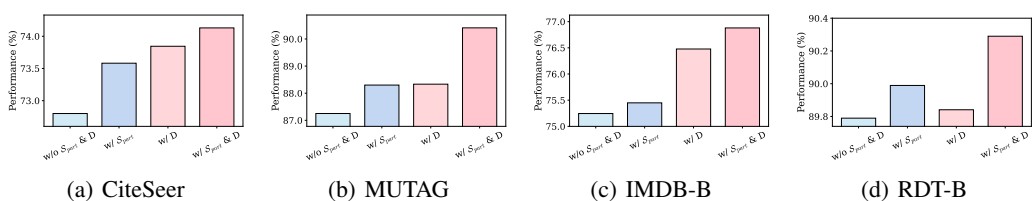

(a) CiteSeer     (b) MUTAG     (c) IMDB-B     (d) RDT-B

Figure 4: Ablation study to analyze the exponential partitioned similarity $\mathcal{S}_{part}$ and the semantics-consistency discriminator $D$.

## B.4 Quantitative analysis

We conduct quantitative analyses to evaluate the effectiveness of the exponential partitioned similarity measure and the semantics-consistency discriminator in CaliGCL.

To assess the partitioned similarity measure, we train the GNN encoder with both the general contrastive loss and the partitioned contrastive loss. We then compare the distribution of positively-

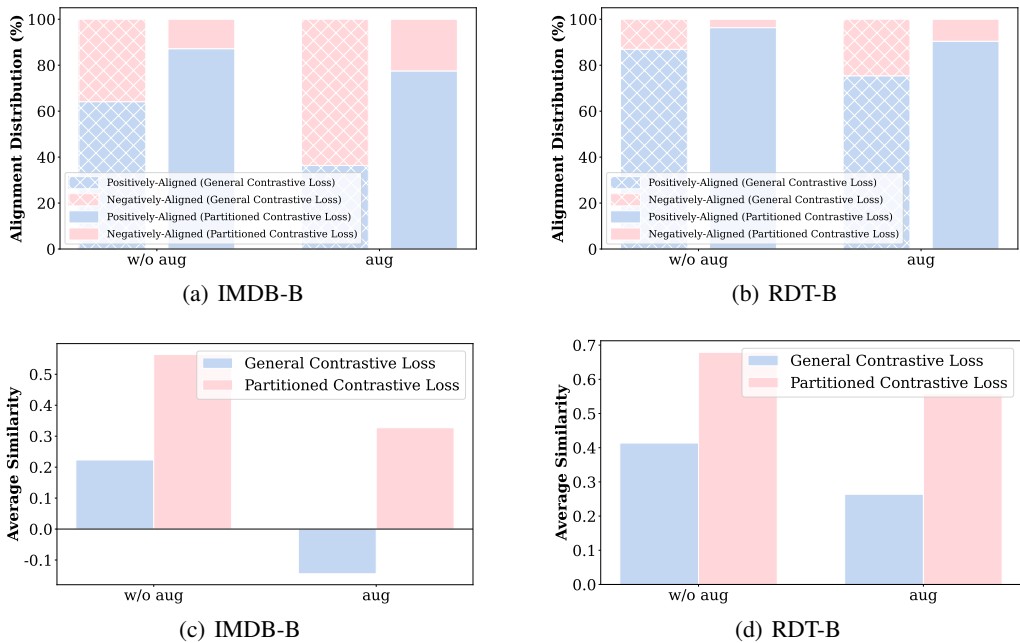

Figure 5: Quantitative analysis on similarity estimation bias, where "aug" means the data with augmentations and "w/o aug" means the data without augmentations.

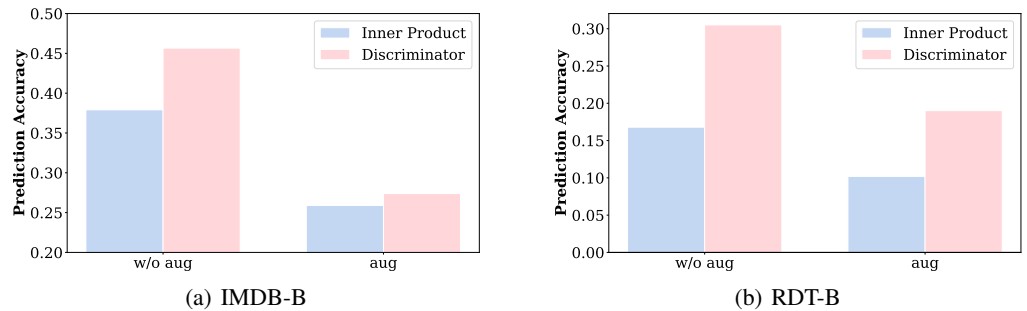

Figure 6: Quantitative analysis on semantic shift bias, where "aug" means the data with augmentations and "w/o aug" means the data without augmentations.

aligned and negatively-aligned partitions in Figure 7(a) and (b) on IMDB-B and RDT-B datasets, as well as the average similarity among positive pairs, which is defined as the total similarity divided by the number of positive pairs in Figure 7(c) and (d). The observed increases in the proportion of positively-aligned partitions and the average similarity clearly demonstrate that the partitioned contrastive loss effectively strengthens positively-synergistic features. Additionally, we compare the average similarity of positives with and without augmentations. Although random augmentations tend to reduce similarity among positives, the graph model using the exponential partitioned similarity still maintains good performance.

To evaluate the function of the semantics-consistency discriminator, we compare its prediction accuracy against similarity-based predictions. Specifically, we count the number of sample pairs predicted as positive pairs by the inner product and by the semantics-consistency discriminator (i.e., pairs with predicted consistency label $\tilde{l} = 1$), and compute their ratios over the ground-truth positives, respectively. As shown in Figure 6, the discriminator yields more accurate predictions, supporting its role in effectively calibrating the contrastive loss.

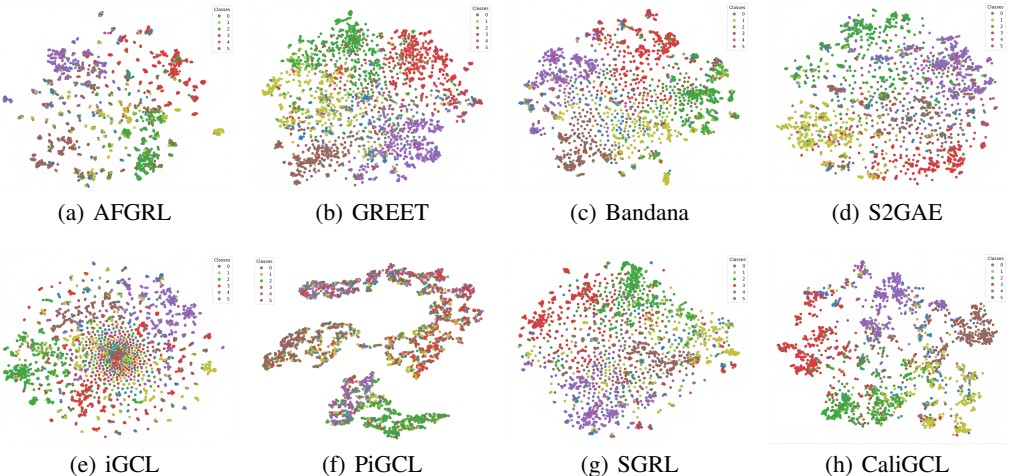

Figure 7: t-SNE visualization of CaliGCL and the comparative models on Citeseer dataset.

## B.5 Visualization

We provide a t-SNE visualization comparing CaliGCL and other baselines to demonstrate the quality of learned representations. From the visualization results, we find the samples of different classes in PiGCL and iGCL are mixed; the boundaries of different classes in GREET, S2GAE, Bandana, and SGRL are ambiguous without margins. In contrast, the boundaries of different classes in CaliGCL are separated with larger margins.

## B.6 Model complexity

We analyze the time and space complexity of CaliGCL by decomposing its key components: (1) Encoder forward pass. For an $L$-layer GNN such as GCN or GIN, the complexity of forward propagation is typically $\mathcal{O}(2L|\mathcal{E}|d)$, where $|\mathcal{E}|$ is the number of edges, $d$ is the representation dimension. (2) Exponential partitioned similarity. For a batch of $N$ nodes, computing similarities with all other $N$ nodes has $\mathcal{O}(KN^2d)$ complexity. (3) Discriminator. For $N^2$ sample pairs, the total complexity is $\mathcal{O}(N^2Dd)$, where $D$ is the dimension of the discriminator. Therefore, the complexity of CaliGCL is $\mathcal{O}(2L|\mathcal{E}|d + (K+D)N^2d)$.

## B.7 Limitation

While the exponential partitioned similarity measure effectively enhances positively-synergistic features and suppresses negatively-synergistic ones, it remains a static mechanism that treats all partitions uniformly without explicitly distinguishing between aligned and misaligned components. A promising future direction would be to develop a learnable or attention-based partition weighting strategy that can automatically identify and emphasize positively-synergistic features while down-weighting negatively-synergistic ones in a data-driven manner.

