# OpenReview forum: "CaliGCL: Calibrated Graph Contrastive Learning via Partitioned Similarity and Consistency Discrimination"
_NeurIPS.cc/2025/Conference — NeurIPS 2025 poster_

### Official Review · Reviewer_8dv3 · 2025-06-20

**Clarity:** 3
**Significance:** 3
**Originality:** 3
**Rating:** 5
**Confidence:** 4

**Summary:**

This paper proposes a novel graph contrastive learning (GCL) model, CaliGCL, which targets at calibrating the similarity estimation bias and the semantic shift bias in the standard GCL framework. Considering this, on one hand, this paper raises an exponential partitioned similarity measure to emphasize the aligned partitions while suppressing conflicting partitions. Besides, the authors also provide theorems to explain the merit of the exponential partitioned similarity measure. On the other hand, this paper designs a semantics-consistency discriminator to assess the semantic relationship between sample pairs, which corrects the contrastive supervision from the semantic shift bias. The model ability is supported through comprehensive experiments on multiple graph datasets against advanced self-supervised graph representation models.

**Questions:**

(1) Does the model training converge? Since balancing the training between the encoder and the discriminator may be difficult, the authors should focus more on the loss convergence to ensure stable model training.
(2) I am curious about whether the block number will cause the index flow? According to Equation 5, a large $K$ will amplify the inner product, which may cause index flow in practice.
(3) The proposed CaliGCL is compared with some self-supervised graph models with learnable augmentations, which optimize the graph augmentations to improve the model performance. However, the paper still states that these models have the potential to distort semantic consistency between sample pairs. So why do these graph models with learnable augmentations still distort semantic consistency between sample pairs, and what are the advantages of CaliGCL when compared with these models?

**Ethical Concerns:**

["NO or VERY MINOR ethics concerns only"]

**Final Justification:**

All my concerns have been addressed and I raise the acceptance ratings.

**Limitations:**

The paper has clarified its limitation and the corresponding resolution, and the authors could refine their resolution to address the limitation.

**Paper Formatting Concerns:**

I have no concerns about the paper formatting.

**Quality:**

3

**Strengths And Weaknesses:**

Strengths:
(1) Writing and organization of this paper are good and logical, and the motivation and the method are clear to understand and follow.
(2) This paper deals with two underexplored bias problems, namely the similarity estimation bias and the semantic shift bias in GCL, and the perspective is interesting.
(3) This paper proposes the exponential similarity measure and the semantics-consistency discriminator to mitigate the two bias problems, which make sense to calibrate the biases in GCL.
(4) Experiments are comprehensive, including two downstream tasks across multiple public graph datasets to demonstrate the model superiority.

Weaknesses:
(1) How to calculate the normalized adjacency matrix $\tilde{A}$ in Line 186 is missing. The authors need to give the specific mathematical formulation of it.
(2) Lines 167-168 state that “the proposed exponential partitioned similarity could exceed the standard exponential inner product across all scenarios”. What do “all scenarios” mean? Why do the authors stress the “all scenarios”?
(3) The authors state that shuffling the feature columns of representations before loss computation could ensure sufficient interactions among feature dimensions across partitions in Lines 220-221. Therefore, it should provide more experimental results to demonstrate the rationality of implementing “Shuffle”.
(4) The authors need to provide a more comprehensive analysis on complexity when compared with other graph models.
(5) The experiments in Figure 6 are conducted to evaluate the discriminator, but the legends are about the loss functions. There is a doubt as to which result was caused by applying the discriminator.

---

> ### Author Rebuttal · Authors · 2025-07-29
>
> **W1**: The normalized adjacency matrix $\tilde{\textbf{A}}$ is calculated as $\tilde{\textbf{A}} = \hat{\textbf{D}}^{-1/2} \hat{\textbf{A}} \hat{\textbf{D}}^{-1/2}$, and $\hat{\textbf{A}}=\textbf{A}+\textbf{I}_N$, $\hat{\textbf{D}}$ is a diagonal matrix, and its diagonal element is ${\hat{\textbf{D}}}\_{ii}={\sum\_j} {\hat{\textbf{A}}\_{ij}}$. This will be clarified in the refined version.
>
> **W2**: “all scenarios” mean all the possible combinations of feature partitions, namely all positively-aligned partitions, all negatively-aligned partitions, and mix of both positively-aligned and negatively-aligned partitions. Lines 167-168 mean that under all possible combinations of feature partitions, exponential partitioned similarity yields similarity greater than or equal to global exponential inner product, where Lines 162-166 only give a simple example to clarify that the proposed exponential partitioned similarity has the ability to amplify the positively-aligned partitions while suppressing the negatively-aligned partitions in the mixed scenario, which cannot generalize to the scenarios where the representation vectors have only positively-aligned or negatively-aligned partitions. And we stress the “all scenarios” to illustrate that through Theorem 2, the proposed exponential partitioned similarity can not only amplify the positively-aligned partitions and suppress the negatively-aligned partitions in the mixed scenario, but can be generalized to common scenarios.
>
> **W3**: Feature shuffling ensures interactions across dimensions and avoids amplifying or suppressing fixed partitions during the training process. To illustrate the function of “Shuffling”, we provide the results of the model with and without shuffling in Tables 1-2. From the results, we could find that the model without shuffling has inferior performance when compared with that with shuffling. It is because that without shuffling, only the fixed partitions are amplified or suppressed during the whole training process. In contrast, the shuffling operator benefits the interactions across feature dimensions and ensures the model performance.
>
> Table 1. Model performance without “Shuffling” for node-level datasets.
>
> |Datasets|Cora|Citeseer|Pubmed|DBLP|Photo|Computers|
> |-|-|-|-|-|-|-|
> |w/o Shuffling|85.58±0.76|73.05±0.71|86.15±0.23|84.50±0.13|92.55±0.21|90.50±0.32|
> |w/ Shuffling|85.87±0.62|74.13±0.54|87.16±0.19|85.32±0.13|93.72±0.24|90.79±0.28|
>
> Table 2. Model performance without “Shuffling” for graph-level datasets.
>
> |Datasets|NCI1|DD|MUTAG|COLLAB|RDT-B|RDT-M5K|IMDB-B|
> |-|-|-|-|-|-|-|-|
> |w/o Shuffling|77.2±1.9|76.7±3.8|90.3±5.3|77.1±1.6|89.5±1.4|56.1±1.6|75.3±5.1|
> |w/ Shuffling|80.8±2.0|80.4±2.5|90.4±5.8|77.2±1.7|90.3±2.7|56.5±1.5|76.9±5.0|
>
> **W4**: To provide a comprehensive comparison among the graph models, we first define $N$ as the number of nodes, $d$ as the dimension of the representations, $D$ as the hidden layer dimension of the discriminator, $|\mathcal{E}|$ as the number of edges,  as the number of sampling adjacent nodes, $C$ as the number of clusters, $V$ as the number of augmented views, $k$ as the number of sampling negatives, $K$ is the number of partitions, and $T$ as the iteration of K-means algorithm. Based on these symbols, we provide the complexity of the comparative models in Table 3. From the results, we find that our model has $\mathcal{O}{(N^2)}$ complexity and though it is more computationally intensive than the graph generative models, it is comparable with the graph contrastive model, which illustrates the practicability of the proposed model.
>
> Table 3. Complexity comparison between graph models.
> |Model|Complexity|
> |-|-|
> |AFGRL|$\mathcal{O}((2L\|\mathcal{E}\|+TNC+N)d+N)$|
> |SUGRL|$\mathcal{O}((L\|\mathcal{E}\|+L+Nm+Nk)d)$|
> |NCLA|$\mathcal{O}((\|\mathcal{E}\|+L\|\mathcal{E}\|+N^2)Vd)$|
> |GREE|$\mathcal{O}(N^{d_s}+(2L\|\mathcal{E}\|+N^2+N^2D)d)$|
> |HomoGCL|$\mathcal{O}((NCT+2L\|\mathcal{E}\|+N^2)d)$|
> |S2GAE|$\mathcal{O}((1+d)L\|\mathcal{E}\|d)$|
> |iGCL|$\mathcal{O}((2L\|\mathcal{E}\|+N^2V+N^2)d)$|
> |PiGCL|$\mathcal{O}(2L\|\mathcal{E}\|d+N^2d+N\log N)$|
> |Bandana|$\mathcal{O}((L+d+Ld)\|\mathcal{E}\|d+\|\mathcal{E}\|)$|
> |SGRL| $\mathcal{O}((2L\|\mathcal{E}\|+\|\mathcal{E}\|+N)d)$|
> |CaliGC|$\mathcal{O}((2L\|\mathcal{E}\|+N^2+N^2D)d)$|
>
> **W5**: It is inappropriate to adopt the loss functions as the legends to cause misunderstandings. In Figure 6, the legend for the blue histogram is the result of the inner product and the legend for the pink histogram is the result of the discriminator. In the refined version, we will replace “General Contrastive Loss” with “Inner Product” and replace “Partitioned Contrastive Loss” with “Discriminator” for clarity.
>
> **Q1**: We have recorded the losses of alternate training strategies (including losses of both encoder and discriminator) during the fine-tuning process to clarify that the model training gradually converges. Besides, both the losses of the encoder and the discriminator become stable in the training process. Since we are unable to provide Figures in the rebuttal, the loss plots will be included in the refined version.
>
> **Q2**: Though $K$ amplifies the magnitude of exponential partitioned similarity, it doesn’t cause index flow in practice. When calculating the contrastive loss with the representations, they are always normalized, which means the inner product of a sample pair is limited in $[-1,1]$. Therefore, the training process will cause index flow unless $K$ is very large. However, according to the experiments, we find $K$ is always smaller than five. Additionally, another hyperparameter, namely the temperature coefficient $\tau$, could also adjust $K$ to avoid the index flow problem.
>
> **Q3**: Though with learnable augmentations, the graph models [1-2] generally optimize the strength of graph augmentations (e.g., the ratio of edge removal), or the combination of adopted augmentation strategies. However, the input data is still modified randomly but with a more appropriate ratio or augmentation combination. Without explicit constraints, they may still destroy meaningful structures or features. In contrast, CaliGCL uses a discriminator to explicitly check semantic consistency between sample pairs and attempts to calibrate the semantics-biased sample pairs to correct the contrastive learning process directly, which further reduces the possibility of semantic shift caused by the random augmentations.
>
> [1]Graph contrastive learning with personalized augmentation, in TKDE 2024.
>
> [2] Adversarial Graph Augmentation to Improve Graph Contrastive Learning, in NeurIPS 2021.

---

> > ### Comment · Reviewer_8dv3 · 2025-08-04
> >
> > Thank you for your responses. All my concerns have been thoroughly addressed, and I appreciate the additional clarifications and experiments provided. Based on the revisions and the strength of the rebuttal, I am updating my score to reflect a more positive evaluation of the paper.

---

> > > ### Author Response · Authors · 2025-08-05
> > >
> > > We sincerely appreciate your positive feedback.

---

### Official Review · Reviewer_7U3j · 2025-06-21

**Clarity:** 3
**Significance:** 4
**Originality:** 4
**Rating:** 5
**Confidence:** 4

**Summary:**

This paper proposes CaliGCL, a graph model for tackling two biases in graph contrastive learning: similarity estimation bias and semantic shift bias. To alleviate the similarity estimation bias, the authors raise an exponential partitioned similarity metric to emphasize positively-aligned partitions and suppress the negatively-aligned partitions. For eliminating the semantic shift bias, the authors apply a semantics-consistency discriminator to calibrate the sample assignments during the contrastive learning process. Empirical results, such as node classification, graph classification, and ablation study demonstrate the ability of CaliGCL to calibrate the biases when compared with other self-supervised graph contrastive and generative models.

**Questions:**

- What do “connected positives” mean in Figure 2? Do the connected positives mean the adjacent nodes? Besides, where are the encoders in Figure 2?

- Why do the authors use the Hadamard product to fuse representations $\z_{v_i}$ and $\z_{v_j}$ but not other fusion strategies like concatenation during the training process of the semantics-consistency discriminator? What is the advantage of the Hadamard product?

- The paper claims that a semantics-consistency discriminator could realize the calibrated sample assignments. Can the authors provide experiment results on whether the discriminator has the ability to identify the positive or negative pairs during training?

- Why don’t the authors sample the negatives and positives with similar sizes when training the discriminator? It can avoid the imbalance problem more efficiently than using a balanced softmax loss.

**Ethical Concerns:**

["NO or VERY MINOR ethics concerns only"]

**Final Justification:**

Thanks for the authors' response, I have no more questions. I will keep my score.

**Limitations:**

The authors have provided some plans for refining the limitations as future work, and the plans are reasonable.

**Paper Formatting Concerns:**

The formatting is OK, I haven’t found any issues.

**Quality:**

3

**Strengths And Weaknesses:**

Strengths:
- This paper deals with the critical problems in graph contrastive learning, namely the similarity estimation bias and the semantic shift bias, which may cause biased representation learning.

- This paper proposes a novel CaliGCL including the exponential partitioned similarity metric and the semantics-consistency discriminator to tackle the two biases and provides the theoretical evidence to analyze the proposed model.

- Extensive empirical experiments on node classification and graph classification show that the proposed CaliGCL is superior to other comparative self-supervised graph models.

Weaknesses:
- The function of doubling the inner product of partitions in the proposed exponential partitioned similarity in Eq. (6) is not intuitive. As the paper claims, the exponential partitioned similarity can suppress the impact of negatively-aligned partitions by inducing their exponentials to approach zero while amplifying positively-aligned ones, it is unnecessary to double the inner product of partitions in Eq. (6). The authors need to clarify why the exponential partitioned similarity still implements the double inner product to achieve the amplification and the suppression.

- Figure 3 only provides the mean performance of the proposed model, it would be better to supplement the standard deviation in Figure 3.

- This paper needs visualization analysis to compare the proposed CaliGCL with other graph models, which better shows the embedding distribution and demonstrates the representation quality.

---

> ### Author Rebuttal · Authors · 2025-07-29
>
> **W1**: Eq. (6) corresponds to an example where a pair of positive samples $(h_v, h_v')$ has both negatively-aligned and positively-aligned partitions, and the inner product of the sample pair is the sum of the inner products of the partitions, namely
>
> $h_{v_i}^T h_{v_j} = [ h_{v_i}^{(1)}]^T [ h_{v_j}^{(1)}]+[h_{v_i}^{(2)}]^T [h_{v_j}^{(2)}].$
>
> Assume that $h_{v_i}^T h_{v_j} > 0$ and the first partition is negatively-aligned, namely $[h_{v_i}^{(1)}]^T [h_{v_j}^{(1)}] < 0$, it is easy to find that the inner product of the second partition has
>
> $[h_{v_i}^{(2)}]^T [h_{v_j}^{(2)}] > h_{v_i}^T h_{v_j}.$
>
> Therefore, as your comment, without doubling the inner product of partitions, it also holds the following conclusion
>
> $\exp\left( \frac{[ h_{v_i}^{(1)}]^T [h_{v_j}^{(1)}]}{\tau} \right) \to 0, \quad \text{and} \quad \exp\left( \frac{[h_{v_i}^{(2)}]^T [h_{v_j}^{(2)} ]}{\tau} \right) \geq \exp\left( \frac{h_{v_i}^T h_{v_j}}{\tau} \right).$
>
> However, this may not hold for the situation where the sample pairs have only positively-aligned or negatively-aligned partitions. For example, when the sample pair is divided into two positively-aligned partitions, we have
>
> $\exp\left( \frac{[h_{v_i}^{(1)}]^T [h_{v_j}^{(1)}]}{\tau} \right) \leq \exp\left( \frac{h_{v_i}^T h_{v_j}}{\tau} \right), \quad \text{and} \quad \exp\left( \frac{[h_{v_i}^{(2)}]^T[h_{v_j}^{(2)}]}{\tau} \right) \leq \exp\left( \frac{h_{v_i}^T h_{v_j}}{\tau} \right).$
>
> It is difficult to compare the values of the original exponential inner product and the sum of the exponential partitioned similarity for judging which is larger. To amplify the sum of the exponential partitioned similarity and make it larger, we follow Jensen’s inequality to leave the original exponential inner product less than or equal to the sum of the exponential partitioned similarity by multiplying the inner product with the number of blocks, and it corresponds to doubling the inner product in Eq. (6). This operator is necessary to generalize the conclusion to more common situations where the partitions are all positively-aligned or negatively-aligned, and we provide the corresponding proof in Theorem 2. Additionally, for Eq. (6), the double inner product has another merit to further facilitate the negatively-aligned partitions to approach zero more quickly while further amplifying the positively-aligned partitions.
>
>
>
> **W2**: We provide the classification results on IMDB-BINARY (IMDB-B) and REDDIT-BINARY (RDT-B) datasets with the means and standard deviations in Tables 1-2. In the refined version, we will include the shaded regions around the broken line in Figure 3 to indicate standard deviations.
>
> Table 1. Hyperparameter analysis for the temperature coefficient.
>
> |Datasets|Tau|Acc|
> |-|-|-|
> |IMDB-B|0.6|76.26±4.80|
> ||0.7|75.68±5.39|
> ||0.8|76.88±5.04|
> ||0.9|75.68±5.33|
> ||1.0|75.48±4.67|
> |RDT-B|1.0|89.44±2.89|
> ||1.1|89.64±2.57|
> ||1.2|90.29±2.69|
> ||1.3|89.69±2.86|
> ||1.4|89.69±2.81|
>
> Table 2. Hyperparameter analysis for the partition size.
>
> |Datasets|Partition Size|Acc|
> |-|-|-|
> |IMDB-BINARY|256|76.48±4.21|
> ||128|76.88±5.04|
> ||86|76.68±4.59|
> ||64|75.87±3.17|
> ||52|75.87±5.04|
> |RDT-B|256|89.84±2.92|
> ||128|90.29±2.69|
> ||86|90.00±2.52|
> ||64|89.69±2.45|
> ||52|89.49±2.51|
>
> **W3**: We provide a t-SNE visualization comparing CaliGCL and other baselines to demonstrate the quality of learned representations. From the visualization results, we find the samples of different classes in PiGCL and iGCL are mixed; the boundaries of different classes in GREET, S2GAE, Bandana, and SGRL are ambiguous without margins. In contrast, the boundaries of different classes in CaliGCL are separated with larger margins. Since we are unable to provide the Figures in the rebuttal, the visualization and the analysis will be included in the refined version.
>
> **Q1**: "Connected positives" refer to nodes that are connected with the anchor node in the topology, and they are the adjacent nodes. Based on the homogeneity principle, these nodes are always considered as positives. The encoder corresponds to the arrows between the input graphs and the feature information, such as feature information $\textbf{H}_U$, $\textbf{H}_V$, and $\textbf{H}\_\mathcal{G}$ in Figure 2. We will supplement the missing details in the refined version.
>
> **Q2**: We use the Hadamard product to preserve fine-grained dimension-wise interactions between node representations. Though both the concatenation and the Hadamard product can preserve the dimension-wise information, the Hadamard product with dimension-wise multiplication emphasizes that the feature elements of a sample pair are closely correlated in corresponding dimensions, which is suited for relationship (similar or dissimilar) prediction. Besides, the Hadamard product doesn’t extend the feature dimensions when compared with the concatenation operator, which will save half the computation cost on calculating the consistency scores than the concatenation operator.
>
> **Q3**: The ability of the discriminator to realize the calibrated sample assignments during training mainly contributes to its identification for semantic consistency of sample pairs. To illustrate the identification ability of the discriminator, we have provided some experiment results to show that the discriminator has superior ability to predict semantic consistency of sample pairs, and the results are shown in Figure 6, where the blue histogram corresponds to the inner product and the pink histogram corresponds to the discriminator.
>
> In the experiments, we use the inner product and the discriminator to judge the semantic consistency of sample pairs. Specifically, we assume the sample pairs with positive inner products are positive in the former method, otherwise negative, while the output of the discriminator can naturally give a judgement of semantic consistency of the sample pairs in the latter method. We compare the predictions with the true labels of the sample pairs to obtain the prediction accuracy, and the results are shown in Figure 6, where the ‘w/o aug’ and ‘aug’ represent the results of the prediction accuracy on the original data and the augmented data, respectively. Figure 6 illustrates that the discriminator has superior performance than the inner product on discriminating the semantic consistency of sample pairs under both the ‘w/o aug’ and ‘aug’ conditions, which demonstrates that the discriminator could further help the model to calibrate the contrastive learning process.
>
> **Q4**: The training for an ideal discriminator always needs numerous positive and negative sample pairs. Since we have no true labels to train the discriminator in an unsupervised setting, we attempt to find the reliable positives and negatives for training the discriminator. In our method, we only pick out the positives from the overlapped samples that are both adjacent and selected with the K-NN algorithm. Based on this strict condition, the positives are more reliable but always very scarce. However, the negative samples are easy to sample, so we can collect much more negative samples than positive samples, which leads to the imbalance problem between the positive and negative sample pairs. This means that though balanced sampling for positives and negatives is more efficient, it is difficult to conduct. To address the inevitable imbalance problem and prevent the encoder from giving consistent predictions as negative pairs, we adopt the balanced softmax loss function instead adjusts the sampling ratio between positives and negatives, offering a more reliable solution to the imbalance problem.

---

> > ### Comment · Reviewer_7U3j · 2025-08-05
> >
> > Thanks for the authors' response, I have no more questions.

---

> > > ### Author Response · Authors · 2025-08-05
> > >
> > > We sincerely appreciate your valuable feedback and we will incorporate all suggestions to improve the final version.

---

### Official Review · Reviewer_cbPN · 2025-06-30

**Clarity:** 2
**Significance:** 3
**Originality:** 3
**Rating:** 4
**Confidence:** 4

**Summary:**

The paper aims to mitigate two problems of graph contrastive learning, \emph{i.e.,} similarity estimation bias and semantic shift bias, which respectively causes truly positive pairs to appear less similar and leads to incorrect positive or negative assignments as well as training noises. Therefore, the authors developed CaliGCL that is a graph contrastive learning model for calibrating the biases by integrating an exponential partitioned similarity measure and a semantics-consistency discriminator. Extensive experiments are performed to demonstrate the effectiveness and superiority of the work.

**Questions:**

1.	While pretraining the discriminator, whether the sample pairs confront the problem of random augmentations?

2.	The paper mentions that “the partition contrastive objective is an upper bound of mutual information than the original contrastive objective”. Generally, the contrastive objective is the lower bound of mutual information. Please provide a more detailed discussion on this issue.

**Ethical Concerns:**

["NO or VERY MINOR ethics concerns only"]

**Final Justification:**

The responses have basically addressed my questions, and I have no further comments and would like to maintain my previous rating.

**Limitations:**

Yes

**Paper Formatting Concerns:**

I did not find much formatting concerns.

**Quality:**

3

**Strengths And Weaknesses:**

Strengths:

1.	This paper provides a deep insight into GCL and introduces two problems faced by existing works;

2.	The proposed solution of exponential partitioned similarity measure is novel, which divides a sample representation vector into multiple partitions. Exponential function is adopted to emphasize the positive partitions and meanwhile it weakens the negative partitions. This method helps to mitigate the degraded similarity between positive samples caused by similarity estimation bias. The semantics-consistency discriminator also helps to distinguish semantically flipped sample pairs.

3.	Extensive experiments are performed to demonstrate the effectiveness and superiority of the work.


Weaknesses:

1.	Some concepts should be further specified, like semantic shift bias and contrastive supervision.

2.	The figures should be further improved. For example, it seems that the augmentations might possibly flip the nature of sample pairs, but Figure 1 only illustrates that positive pairs might become negative pairs. How about that the negative samples become positive pairs? Similarly, Figure 2 should include more details.

3.	The design of discriminator is somehow unclear.

---

> ### Author Rebuttal · Authors · 2025-07-29
>
> **W1**: Semantic shift bias corresponds to the semantic relationships between sample pairs which are altered due to augmentations, such as the positive pairs becoming negative pairs. The contrastive supervision is the pre-defined supervision about how to realize the positive or negative assignments. In graph contrastive learning, the contrastive supervision is to assign corresponding nodes across views as positives while others as negatives.
>
> **W2**: Figure 1 shows that with augmentations, the ratio of positively-aligned partitions decreases, indicating that some positive pairs are no longer aligned and supporting the existence of semantic shifts. However, Figure 1 focuses on the condition that a part of positive pairs become negative pairs. To illustrate the condition that the negative pairs may become positive pairs, we count the negative pairs whose inner product similarities become positive after augmentation, and find that 99,742 negative pairs show semantic shifts. This illustrates the possibility that negative pairs turn into latent positive pairs. The relevant statistics will be added in Figure 1 in the refined version.
>
> For Figure 2, we will supplement these semantics-bias pairs after the augmentation process in detail in the refined manuscript.
>
> **W3**: Since the augmentations may change the semantic consistency between sample pairs, we propose the discriminator designed for finding the semantic-consistent pairs for calibrating the contrastive process and preventing semantic shift bias. And it is designed as a four-layer MLP with activation functions. We provide the specific configurations across different datasets in Tables 1-2, where Dis_dim is the hidden dimension of the discriminator and Act is the activation function applied in the hidden layer. The last layer will add a Sigmoid function to output the consistency scores.
>
> Table 1. Design of the discriminator for node-level datasets.
>
> |Datasets|Cora|Citeseer|Pubmed|DBLP|Photo|Computers|
> |-|-|-|-|-|-|-|
> |Dis_dim|128|512|256|128|128|512|
> |Act|LeakyReLU|LeakyReLU|LeakyReLU|LeakyReLU|LeakyReLU|LeakyReLU|
>
> Table 2. Design of the discriminator for graph-level datasets.
>
> |Datasets|MUTAG|IMDB-B|COLLAB|RDT-B|DD|NCI1|RDT-M5K|
> |-|-|-|-|-|-|-|-|
> |Dis_dim|128|128|128|128|64|64|128|
> |Act|LeakyReLU|LeakyReLU|LeakyReLU|LeakyReLU|LeakyReLU|LeakyReLU|LeakyReLU|
>
>
> **Q1**: The sample pairs used for pre-training have not undergone random augmentations, since the random augmentations may change the semantic consistency of sample pairs. When using the biased samples to train the discriminator, it must restrict its reliability and effectiveness. In order to pick out reasonable positive and negative pairs, we only select the positive and negative pairs in the original graph based on the K-NN algorithm and the homogeneity principle.
>
>
> **Q2**: The same in your comment, both the partition contrastive objective and the original contrastive objective follow the InfoNCE objective form that is the lower bound of the mutual information. Theorem 3 is proposed for proving that the proposed partition contrastive objective provides a tighter upper bound than the contrastive objective, not an upper bound of mutual information. We provide discussions on this issue in the following:
> Formally, the InfoNCE objective for a positive pair $(h, h_+)$ over a batch of size $N$ is
>
> $\mathcal{J}\_{\text{Info}} = \log \left( \frac{\exp(S(h, h_+)/\tau)}{\sum_h \exp(S(h, h)/\tau)} \right)$
>
> It corresponds to
>
> $I(h; h_+) \geq \log N + \mathcal{J}_{\text{Info}}.$
>
> Therefore, the InfoNCE objective is the lower bound of the mutual information. In the paper, the partitioned exponential similarity is defined as:
>
> $S_{\text{part}}(h, h_+) = \frac{1}{K} \sum_k \exp \left( \frac{K \cdot \left[ h^{(k)} \right]^T \left[ h_+^{(k)} \right]}{\tau} \right),$
>
> where $h^{(k)}$ and $h_+^{(k)}$ are the $k$-th partitions of representations $h$ and $h_+$, respectively. The partitioned exponential similarity results in the partitioned contrastive objective as
>
> $\mathcal{J}\_{\text{part}} = \log \left( \frac{S_{\text{part}}(h, h_+)}{\sum_h S_{\text{part}}(h, h_+)} \right).$
>
> Since the objective obeys the InfoNCE formation, we also have
>
> $I(h; h_+) \geq \log N + \mathcal{J}_{\text{part}}.$
>
> Hence, both the contrastive objective and the proposed objective are the lower bounds of the mutual information. However, Theorem 3 proves $\mathcal{J}_{\text{part}}$ is the upper bound of $\mathcal{J}\_{\text{Info}}$, which leads to the following conclusion:
>
> $I(h; h_+) \geq \log N + \mathcal{J}\_{\text{part}} \geq \log N + \mathcal{J}\_{\text{Info}}.$
>
> It means $\mathcal{J}\_{\text{part}}$ is a more ideal objective than $\mathcal{J}\_{\text{Info}}$, which induces a tighter lower bound on mutual information. Theorem 3 provides the inappropriate wording for misunderstandings. We will revise the wording of Theorem 3 as “The partition contrastive objective serves as a tighter lower bound on mutual information, and thus yields an upper-bounded estimate compared to the contrastive objective” in the refined version.

---

> > ### Comment · Reviewer_cbPN · 2025-08-09
> >
> > The responses have basically addressed my questions, and I have no further comments and would like to maintain my previous rating.

---

### Official Review · Reviewer_hzBf · 2025-07-01

**Clarity:** 3
**Significance:** 2
**Originality:** 3
**Rating:** 4
**Confidence:** 4

**Summary:**

This paper proposes CaliGCL, a calibrated contrastive learning framework for graph representation learning. The method adjusts the negative sample distribution in contrastive loss using a soft calibration mechanism based on anchor hardness and positive similarity. Experiments on five benchmarks show that CaliGCL improves performance over various unsupervised and supervised GCL baselines. The authors also provide theoretical insights into calibration effects on similarity distributions.

**Questions:**

-Could the authors provide a sensitivity analysis regarding the calibration function components (e.g., anchor hardness and positive similarity)? It is unclear how each individually contributes to performance gains.

-The theoretical section, particularly Equation (4) and the definition of entropy-based hardness, lacks clarity. Could the authors elaborate on the derivation and its role in model behavior?

-How does CaliGCL scale in terms of time and memory complexity on large-scale graphs? An empirical or analytical comparison would be helpful to assess real-world applicability.

-How does CaliGCL fundamentally differ from existing adaptive weighting or calibrated contrastive loss approaches in vision or graph domains? Please clarify the novel contribution beyond reweighting strategies.

**Ethical Concerns:**

["NO or VERY MINOR ethics concerns only"]

**Final Justification:**

The authors have addressed all of my concerns. After also reviewing the comments and perspectives from other reviewers, I have decided to raise my score to a 4.

**Limitations:**

yes

**Quality:**

3

**Strengths And Weaknesses:**

Strengths：

-The paper targets a relevant and timely problem in GCL, namely the lack of semantic awareness in existing negative sampling strategies.

-The proposed calibration mechanism is conceptually intuitive and can be incorporated into standard contrastive objectives with minimal modification.

-Experimental results show consistent improvements on well-known benchmarks.

Weaknesses：

-Several parts of the theoretical analysis (e.g., Eq. 4, definitions of entropy and hardness) are not clearly explained, making the formal contribution hard to interpret for a broader audience.

-The effect of individual components in the calibration strategy (e.g., hardness vs. positive similarity) is not sufficiently analyzed. This makes it hard to assess where the gains truly come from.

-The paper does not discuss how well the method scales to large graphs or more realistic industrial settings, which limits its practical relevance.

-While the calibration idea is useful, it can be seen as an extension of existing weighting schemes in contrastive learning, with limited theoretical or algorithmic innovation.

---

> ### Author Rebuttal · Authors · 2025-07-30
>
> ***
> We have carefully read your comments and made extensive efforts to understand your concerns, particularly the core issue regarding the definitions of “entropy” and “hardness”.
>
> At the beginning, we think that the concepts of entropy and hardness you mentioned might not be directly applicable to our model, since our motivation is to calibrate biased similarity estimation caused by conflicting feature elements and address semantic inconsistency arising from data augmentations but doesn’t explicitly incorporate entropy-based reweighting or hardness-aware mining in the traditional sense.
>
> However, we also reconsider what deeper concern or theoretical ambiguity might have motivated your comment and why these concepts are brought up. Considering this, we attempt to infer your original intent and the underlying motivation behind your question, and suppose that “entropy” denotes the softmax-like contrastive loss, and “hardness” is the trade-off between a learning signal from hard negatives and the harm due to the correction of false negatives. Guided by this understanding, we provide the following response in hopes that it can address your concerns.
>
> If we misunderstand your question or don't resolve the issues you intend to highlight, we would greatly appreciate any clarification in the discussion phase and try our best to provide a more targeted and clear response.
> ***
>
> **W1 & Q2**: We would like to clarify that Eq. 4 does not involve this issue, and we assume that the equation you referred to might be Eq. 5, and our responses are based on Eq. 5.
>
> “Entropy” denotes the softmax-like contrastive loss, such as cross-entropy [1-2]. Given the fact that the contrastive loss is regarded as the cross-entropy, the definition of entropy is obtained by substituting Eq. 5 into Eq. 10 as
>
> $\text{Entropy} = - \log \left( \frac{\sum_k \exp \left( K[h_u^{(k)}]^T h_v^{(k)} \right) / \tau}{\sum_j \sum_k \exp \left( K[h_u^{(k)}]^T h_{v_j}^{(k)} \right) / \tau + \sum_{j \neq i} \sum_k \exp \left( K[h_u^{(k)}]^T h_{u_j}^{(k)} \right) / \tau} \right)$
>
> where $K$ is the number of partitions, $\tau$ is the temperature, $u_i$ and $v_i$ are the $i$-th samples in the augmented views $U$ and $V$, respectively. Here, $(u_i, v_i)$ is a positive pair, $(u_i, v_j)$ and $(u_i, u_j), j \neq i$ are negative pairs.
>
> “Hardness” reflects the trade-off between learning from difficult negatives and avoiding false negatives [3-4]. In our unsupervised setting without labels, this difficulty arises from distinguishing positive and negative pairs. To address this, we introduce a learnable semantics-consistency discriminator that predicts whether a pair is positive or negative based on a threshold $\eta$. We then define hardness as the absolute difference between the predicted consistency score and the threshold, rather than using an entropy-based formulation as
>
> $\text{hardness} = 1 - |p_{v_i, v_j} - \eta|$
>
> where $p_{v_i, v_j}$ is the predicted consistency score of pair $(v_i, v_j)$, and $\eta$ is the threshold for judging whether the sample pairs are positive or negative.
>
> In the theoretical section, Eq. (5) defines an enhanced contrastive similarity designed to reduce the influence of conflicting features within representation vectors. Traditional inner product similarity may suppress positively-synergistic signals due to cancellation from negatively-synergistic ones, even within positive pairs. To address this, Theorems 2 and 3 illustrate the behavioral benefits:
>
> 1. Theorem 2 shows that the exponential partitioned similarity in Eq. (5) upper-bounds the standard exponential inner product. This allows positive pairs with aligned partitions to receive higher similarity scores, even when negatively-aligned partitions would otherwise suppress them, thereby enhancing contrastive discriminability.
>
> 2. Theorem 3 shows that using partitioned similarity leads to a contrastive loss that provides a tighter lower bound on mutual information. This implies that our method better preserves semantic consistency and promotes the learning of more informative representations.
>
> These theorems provide both mathematical support and behavioral interpretation for our model.
>
> [1] Representation Learning with Contrastive Predictive Coding, in Arxiv 2018.
>
> [2] A Simple Framework for Contrastive Learning of Visual Representations, in ICML 2020.
>
> [3] Understanding the Behaviour of Contrastive Loss, in CVPR 2021.
>
> [4] Debiased Contrastive Learning, in NeurIPS 2020.
>
> **W2 & Q1**: We conduct experiments to show the contributions of the anchor hardness or the positive similarity in the calibration strategy. To evaluate the anchor hardness, we remove hard negatives of anchors (hardness $\ge$  0.9 and consistency score < $\eta$) from the loss. Similarly, to evaluate the positive similarity, we remove all the positive pairs (consistency score  $\ge \eta$) but preserve the corresponding samples of anchors in the augmented views in the loss. The results are listed in Table 1. Additionally, we provide the performance gaps between CaliGCL and the models without either anchor hardness or positive similarity. Generally, larger gaps correspond to greater contributions of the components. From the results, we find that both the anchor hardness and positive similarity contribute to the performance gains, which validates the effectiveness of the discriminator to calibrate the false-negatives and false-positives. Besides, we observe that positive similarity contributes more in small datasets (Citeseer, Photo, and Computers) while the anchor hardness contributes more in relatively large datasets (Pubmed and DBLP).
>
> Table 1. Comparison of performance gains concerning anchor hardness and positive similarity.
>
> |Datasets|Cora|Citeseer|Photo|Computers|DBLP|Pubmed|
> |-|-|-|-|-|-|-|
> |w/o anchor hardness|85.76|73.77|93.46|90.64|84.42|86.17|
> |w/o positive similarity|85.79|73.61|93.44|90.50|84.65|86.26|
> |CaliGCL|85.87|74.13|93.72 |90.79|85.32|87.16|
> |Performance Gap (without anchor hardness/positive similarity)|**0.11**/0.08|0.36/**0.52**|0.26/**0.28**|0.15/**0.29**|**0.90**/0.67|**0.99**/0.90|
>
> **W3 & Q3**: We provide the comparative results on model performance, training time, and GPU memory across two large-scale graph datasets, namely Arxiv and Mag in Table 2. The results demonstrate that CaliGCL can be applied to the large-scale datasets. For the model performance, we could observe that CaliGCL has the best performance when compared with other graph models across the two large-scale datasets, which illustrates that the calibration strategy aids in refining the contrastive process and enhances the model performance. However, the training time and GPU memory are heavier than other models due to the discrimination process. Even so, the time and space overheads of the model are within several times that of other models, and overall, they are at the same level.
>
> Table 2. Performance and complexity comparison among the graph models across large-scale datasets.
> |Datasets|Arxiv|||Mag|||
> |-|-|-|-|-|-|-|
> ||Acc (%)|Time (s)|GPU (M)|Acc (%)|Time (s)|GPU (M)|
> |AFGRL|56.14±0.14|1308.15| 6966.99|27.97±0.02|5795.03|8370.58|
> |NCLA|58.43±0.02|999.17|**2041.58**|25.29±0.01|3470.89|**2887.58**|
> |HomoGCL|62.53±0.48|**566.54**|12194.19|26.17±0.02|**3197.28**|18131.88|
> |S2GAE|60.88±0.00|1303.80|11455.05|23.66±0.00|4374.16|14795.73|
> |Bandana|66.78±0.00|655.88|9947.60|25.91±0.00|5003.85|15452.42|
> |SGRL|65.71±0.01|673.02|14027.77|26.68±0.33|7506.63|24060.54|
> |CaliGCL|**68.42±0.00**|1969.10|14853.70|**28.66±0.00**|11126.59|23586.28|
>
>
> **W4 & Q4**: We clarify that CaliGCL differs from existing adaptive weighting methods [1-3] in terms of motivation, methodology, and experiment results. Specifically, we highlight the following key contributions or innovations from theoretical and algorithmic perspectives:
>
> **Algorithmic Innovations:**
>
> 1. Broader bias correction: While most existing GCL methods focus on false negatives [1–2], they often overlook false positive cases where augmentations disrupt true positive pairs. CaliGCL’s discriminator addresses both, enabling more comprehensive correction of contrastive supervision.
>
> 2. Comprehensive calibration perspective: Previous adaptive methods typically rely on feature similarity [1] or homophily principle [2] for reweighting. Except for them, CaliGCL introduces a finer-grained partition-level view, allowing calibration at a more detailed level beyond feature similarity or local adjacency.
>
> 3. Learnable semantics-consistency: Unlike prior methods that rely on fixed or heuristic similarity measures [1–3], CaliGCL uses a learnable discriminator to adaptively assess whether augmented pairs remain semantically consistent. This leads to more accurate supervision, especially in complex scenarios.
>
> 4. Empirical superiority: Beyond theoretical contributions, CaliGCL consistently outperforms existing adaptive weighting methods across various datasets, demonstrating its effectiveness in addressing semantic shifts and feature conflicts for better representation learning.
>
> **Theoretical Innovations:**
>
> The theoretical section provides a solid foundation for why our method generates more expressive representations than existing methods.
>
> 1. Theorem 1 reveals the existence and effect of negatively-aligned partitions in positive pairs, supporting our motivation to suppress their influence.
>
> 2. Theorem 2 proves that the proposed exponential partitioned similarity yields a more ideal similarity than the inner product, thereby improving feature discrimination.
>
> 3. Theorem 3 further shows that the partition contrastive loss serves as a tighter upper bound on the mutual information lower bound, meaning that it preserves or improves representation quality than the standard contrastive loss.
>
> [1] HomoGCL: Rethinking Homophily in Graph Contrastive Learning, in KDD 2023.
>
> [2] Enhancing Contrastive Learning on Graphs with Node Similarity, in KDD 2024.
>
> [3] Debiased Contrastive Learning, in NeurIPS 2020.

---

> > ### Comment · Reviewer_hzBf · 2025-08-07
> >
> > Thank you for your detailed and constructive response. Based on your response and the additional insights, I have decided to increase my rating.

---

> > > ### Author Response · Authors · 2025-08-07
> > >
> > > Thank you very much for your kind constructive feedback. We truly appreciate your thoughtful response and are glad to hear that our clarifications were helpful. We will incorporate your valuable suggestions into the final version of the paper to further enhance its quality.

---

> ### Author Response · Authors · 2025-08-06
>
> Dear Reviewer,
>
> We sincerely appreciate your time and insightful comments on our paper. As the author-reviewer discussion phase is nearing its deadline, we’d like to kindly confirm if you’ve had a chance to review our responses or if you require any additional clarifications from our side.
>
> Your feedback is crucial to our work, and we’d be grateful for any final thoughts you may have before the discussion period closes.
>
> Thank you again for your efforts!
>
> Best regards,
>
> Authors.

---

### Decision · Program_Chairs · 2025-09-17

**Decision:**

Accept (poster)

**Comment:**

This paper introduces a calibrated contrastive learning framework for graph representation learning. CaliGCL adjusts the negative sample distribution with a soft calibration mechanism based on anchor hardness and positive similarity. Both theoretical and empirical results are provided to demonstrate the effectiveness. The writing clarity is argued by multiple reviewers, and has been alleviated after the rebuttal. Other minor issues, such as more in-depth analysis of experimental results or the exploration on large-scale graphs, are also suggested for improvement. All four reviewers give positive scores to this submission. Considering the scope of this paper, I will recommend acceptance as a poster.